# The critical role of SENP1-mediated GATA2 deSUMOylation in promoting endothelial activation in graft arteriosclerosis

Cong Qiu[1,2], Yuewen Wang[1,2], Haige Zhao[3], Lingfeng Qin[4,5], Yanna Shi[1,2], Xiaolong Zhu[1,2], Lin Song[1,2], Xiaofei Zhou[1,2], Jian Chen[1], Hong Zhou[1], Haifeng Zhang[4,5], George Tellides[4,5], Wang Min[4,5,6] & Luyang Yu[1,2]

Data from clinical research and our previous study have suggested the potential involvement of SENP1, the major protease of post-translational SUMOylation, in cardiovascular disorders. Here, we investigate the role of SENP1-mediated SUMOylation in graft arteriosclerosis (GA), the major cause of allograft failure. We observe an endothelial-specific induction of SENP1 and GATA2 in clinical graft rejection specimens that show endothelial activation-mediated vascular remodelling. In mouse aorta transplantation GA models, endothelial-specific SENP1 knockout grafts demonstrate limited neointima formation with attenuated leukocyte recruitment, resulting from diminished induction of adhesion molecules in the graft endothelium due to increased GATA2 SUMOylation. Mechanistically, inflammation-induced SENP1 promotes the deSUMOylation of GATA2 and IκBα in endothelial cells, resulting in increased GATA2 stability, promoter-binding capability and NF-κB activity, which leads to augmented endothelial activation and inflammation. Therefore, upon inflammation, endothelial SENP1-mediated SUMOylation drives GA by regulating the synergistic effect of GATA2 and NF-κB and consequent endothelial dysfunction.

[1] College of Life Sciences, Institute of Genetics and Regenerative Biology, Zhejiang University, Hangzhou, Zhejiang 310058, China. [2] Research Center for Air Pollution and Health, Zhejiang University, Hangzhou, Zhejiang 310058, China. [3] Department of Cardiothoracic Surgery, First Affiliated Hospital, School of Medicine, Zhejiang University, Hangzhou, Zhejiang 310058, China. [4] Interdepartmental Program in Vascular Biology and Therapeutics, Yale University School of Medicine, New Haven, Connecticut 06520, USA. [5] Department of Pathology and Surgery, Yale University School of Medicine, New Haven, Connecticut 06520, USA. [6] The First Affiliated Hospital, Center for Translational Medicine, Sun Yat-sen University, Guangzhou 510080, China. Correspondence and requests for materials should be addressed to L.Y. (email: luyangyu@zju.edu.cn).

Organ transplantation is the most effective treatment for end-stage organ failure, but its success is hindered by immune-mediated rejection. Although current immuno-suppressive regimens work well for acute graft rejection, they have limited effects on chronic rejection, which results in substantial morbidity of the graft and even death in 90% of recipients within 1–10 years post-transplantation[1]. Pathologically, arteries within graft organs bear the brunt of the recipient immune system, and the resultant graft arteriosclerosis (GA) is the main contributor to vascular occlusion and the consequent graft ischaemic failure[2,3]. Therefore, a better understanding of GA is essential for the prevention and treatment of organ transplant failure.

GA is an arteriosclerotic condition characterized by intimal thickening and vasoregulatory dysfunction, which arises from localized and restricted leukocyte recruitment to the graft endothelium. Driven by local pro-inflammatory cytokines, GA develops as prototypic vascular remodelling characterized by occlusive neointima composed of accumulated immune cells and vascular smooth muscle cells (VSMCs). Although both GA and atherosclerosis are characterized by intima hyperplasia with immune cell and VSMC involvement, they are different vascular diseases based on their pathological characteristics. GA proceeds rapidly in extensive vessels, covering most of the arterial vasculature, and develops diffused and concentric intimal thickening. By contrast, atherosclerosis shows a slower progression and affects only major epicardial muscular arteries, developing focal fibrofatty plaques mostly with a necrotic core and fibrous cap[4].

Although multiple factors, such as human leukocyte antigens, ischaemic stress and surgical injury, have been identified as stimuli for graft-associated vasculopathy[1], the initiation of GA is attributed to endothelial phenotypic switch[5]. During host immune responses, the graft arterial endothelium is activated by pro-inflammatory cytokines and highly expresses cell adhesion molecules including E-selectin, P-selectin, intercellular adhesion molecule (ICAM)-1, vascular cell adhesion molecule (VCAM)-1 and platelet/endothelial cell adhesion molecule (PECAM)-1. Induced endothelial adhesion molecules establish a requisite environment for early adhesion and subsequent transmigration of inflammatory cells into the vessel wall followed by migration and proliferation of VSMCs, resulting in vascular remodelling[6,7]. Upregulation of adhesion molecules involves transcriptional mechanisms in which the importance of the GATA family has been well recognized[8–10]. Among six GATA transcription factors (GATA1 to GATA6), previous studies have demonstrated that GATA2 is essential for cardiovascular development in addition to its role in haematopoietic stem cells[11–13]. GATA2 is abundantly expressed in vascular endothelial cells (ECs)[14–16] and has been identified as a key transcription factor that regulates endothelial-specific genes, such as VCAM-1, P-selectin and PECAM-1 (refs 17–19), which are classic markers of EC activation. Although accumulating evidence supports the association between GATA2 and EC activity, it is unclear whether GATA2 is involved in GA and vascular remodelling.

Protein post-translational modifications, such as phosphorylation, acetylation, methylation, ubiquitination and small ubiquitin-like modifier (SUMO) modification (or SUMOylation), have been shown to play important roles in altering the physiological states of cells. The SUMO family is composed of SUMO1–3, in which SUMO2 and SUMO3 share 95% sequence identity and have identical functions[20]. Similar to ubiquitination, SUMOylation is characterized by the covalent conjugation of SUMO to substrates, which is mediated by activating (E1), conjugating (E2) and ligating (E3) enzymes[21,22]. Nuclear proteins, especially transcription factors, are a major group of SUMOylation targets. SUMOylation is not generally related to elevated protein degradation, but it broadly affects a number of processes involving its target proteins, including cellular localization, protein stability, and protein–protein or protein–DNA interactions[23–29]. This difference may be due to SUMO-substrate conjugations promoted by E3 ligase and to deconjugation by SUMO-specific protease (SENP), which are different from those involved in ubiquitination. As one of the most dynamic modifications, SUMO molecules are readily decon-jugated by six SENP family proteins: SENP 1–3 and SENP 5–7 (refs 24–27,30). Among SENPs, SENP1 is largely responsible for the deconjugation of both SUMO1 and SUMO2/3 modifications in a large number of target proteins and is, thus, involved in many cellular processes[25]. So far, SENP1 has been shown to regulate the activity of dozens of transcription factors in vitro. Previous studies, including our own work, have revealed the pathophysiological role of SENP1 in haematopoiesis using SENP1 conventional knockout (KO) mice[31,32]. GATA1 and HIF1α, two common transcription factors, were identified as targets of SENP1. Interestingly, our data also indicated the potential correlation between SENP1 deficiency and vascular defects[31]. However, the role of SENP1 in the vascular system has hardly been investigated.

Clinical studies have suggested the important roles of SUMOylation in cardiovascular disease. Abnormal expression of SUMO system proteins was observed in various vascular cell types such as ECs, indicating that ECs could be the major target of SUMOylation. In recent studies, shear stress-induced endothe-lial SUMOylation has been found to regulate ERK5 and p53, suggesting an important role for SUMOylation in EC dysfunction and atherosclerosis[33–35]. However, the potential role of SUMOylation in GA development is currently unexplored. GATA2 function is tightly regulated by several post-translational modifications, including phosphorylation, acetylation and ubiqui-tination. Notably, one previous study indicated that GATA2 transcriptional activity in ECs was directly modified by the SUMO E3 ligase PIASy in vitro[11], which implies the possibility of GATA2 SUMOylation in ECs. In the present study, we demonstrate that the loss of endothelial SENP1 prevents EC activation and subsequent GA development as a result of enhanced SUMOylation of GATA2 and IκBα, which reduces adhesion molecule induction. Our results reveal a mechanism whereby endothelial inflammation is modulated by SENP1-mediated SUMOylation during GA.

## Results

**Expression of endothelial SENP1 and GATA2 correlates with GA.** A potential role for SENP1 in vascular defects was suggested in our previous study[31]. To determine the expression pattern of SENP1 in clinical GA, similarly sized human coronary arteries from transplanted hearts with GA and from non-diseased hearts were collected for histological examination. SENP1 was weakly expressed in non-diseased vessels, whereas its expression was dramatically enhanced in the luminal endothelial layer (CD31$^+$) of rejecting arteries; the changes were proportional to the severity of rejection (Fig. 1a,b). Elastic-Van Gieson staining showed obvious intimal expansion with abundant CD45$^+$ inflammatory cell infiltration (Supplementary Fig. 1A–C). Thus, to further determine a link between endothelial SENP1 expression and vascular inflammation, VCAM-1 and P-selectin expression was assessed by immunofluorescence staining. As markers of EC activation, both VCAM-1 and P-selectin expression was markedly enhanced in the endothelium of the rejecting arteries compared with that of non-diseased controls, which is consistent with the pattern of SENP1 expression and inflammation within the vessel wall (Fig. 1c). Intriguingly, endothelial expression of GATA-2, a major transcription factor in EC activation, was elevated during

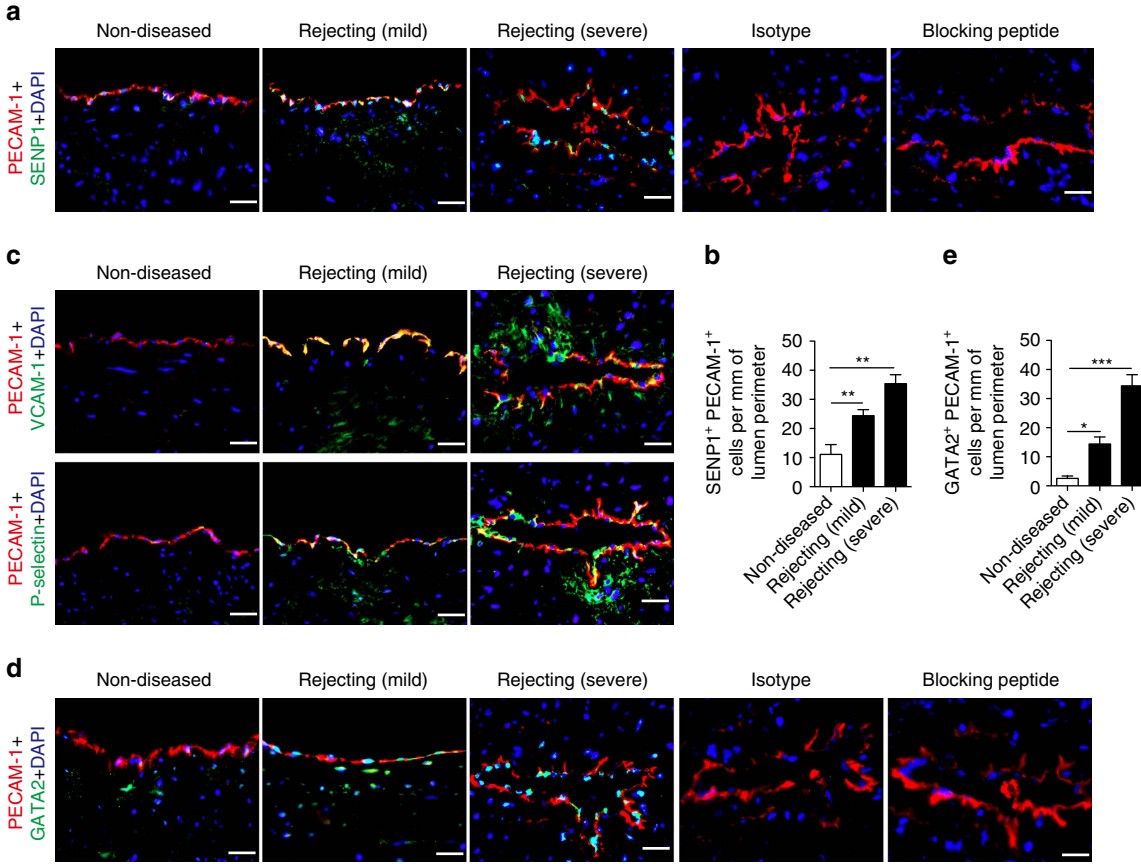

**Figure 1 | Enhanced expression of endothelial SENP1 and GATA2 correlates with graft arteriosclerosis (GA) progression.** Similarly sized human coronary arteries with GA from chronically rejecting heart allografts or without disease from non-transplanted hearts were collected and evaluated by histological analysis. (**a,b**) Dramatically increased expression of endothelial SENP1 was detected in the diseased vessel wall. Endothelial SENP1 expression is demonstrated by immunofluorescence analysis of coronary artery cross-sections that were stained for SENP1 and the endothelial marker PECAM-1 with DAPI labelling of the nuclei. Representative images are shown in (**a**) with quantification data in (**b**). Bar represents 50 μm. (**c**) Induction of endothelial adhesion molecules resulted in a similar augmented pattern as endothelial SENP1. Representative images of immunofluorescence staining for VCAM-1 or P-selectin and PECAM-1 in coronary arteries with DAPI counterstaining are shown. Bar represents 50 μm. (**d,e**) Expression of endothelial GATA2 was elevated with the progression of aggravated rejection. Representative images of the immunofluorescence staining of coronary arteries for GATA2 and PECAM-1 together with DAPI nuclear staining are shown in (**d**) with quantification data in (**e**). Negative SENP1 and GATA2 staining in the isotype or blocking peptide controls are also shown in (**a**) and (**d**). Bar represents 50 μm. Data presented in (**b,e**) are the mean ± s.e.m. from five separate clinical specimens per group as indicated. *$P < 0.05$, **$P < 0.01$ and ***$P < 0.0001$; one-way ANOVA followed by Bonferroni test. GA, graft arteriosclerosis; PECAM, platelet/endothelial cell adhesion molecule; SENP, sentrin-specific protease; VCAM, vascular cell adhesion molecule.

rejection progression (Fig. 1d,e). These results suggest a positive role for SENP1 in EC activation and vascular inflammation during pathological vascular remodelling in GA.

**Vascular endothelial deletion of SENP1 in mice inhibits GA.** To investigate the role of endothelial SENP1 in GA, we generated vascular EC-specific SENP1 knockout mice (SENP1-ecKO) by mating SENP1 flox (*SENP1 flox/flox*) mice[31] with vascular EC-specific VE-cad-Cre mice. Both WT and SENP1-ecKO mice were backcrossed onto a C57BL/6 (B6) background, and they were viable and fertile. EC-specific SENP1 knockout was verified by western blotting in primary mouse aortic endothelial cells (MAECs) from WT and SENP1-ecKO mice and by immuno-fluorescence staining in aortas. The knockout efficiency was sufficient, as indicated in Supplementary Fig. 2A,B. Neither vessel morphology differences (Fig. 2a, columns 1 and 3) nor vessel function variation (Supplementary Fig. 2C–E) was observed in the aortas of WT and SENP1-ecKO mice.

The pathological role of endothelial SENP1 was evaluated in a mouse aorta transplantation model of GA that was established in

our previous study[36]. The B6 female host mounted an alloimmune response against the male-specific H-Y minor histo-compatibility antigen, whereas the male host showed no rejection to the male donor tissue (Fig. 2a, columns 1 and 2). Consistent with the observations in clinical samples, enhanced expression of endothelial SENP1 was detected in the WT graft compared to the rejecting graft (Supplementary Fig. 3). At 2 weeks post-transplantation, donor aortas were harvested for histological analysis and morphometric assessment. WT male to B6 female aorta transplantation-induced GA was characterized by graft infiltration by leukocytes and neointima formation. By contrast, SENP1-ecKO grafts generated significantly less neointima that contained fewer CD45[+] inflammatory cells, thus maintaining a better lumen size (Fig. 2a,f with quantification in Fig. 2b–e and Fig. 2g). Compared with the male-to-male graft that exhibits no obvious vascular remodelling and intimal hyperplasia, SENP1-ecKO male-to-female grafts exhibit a similar lumen area but have minor neointima formation, whereas the WT male-to-female graft exhibits reduced lumen area with considerable neointima formation (Fig. 2a, columns 2 and 4 compared with column 1).

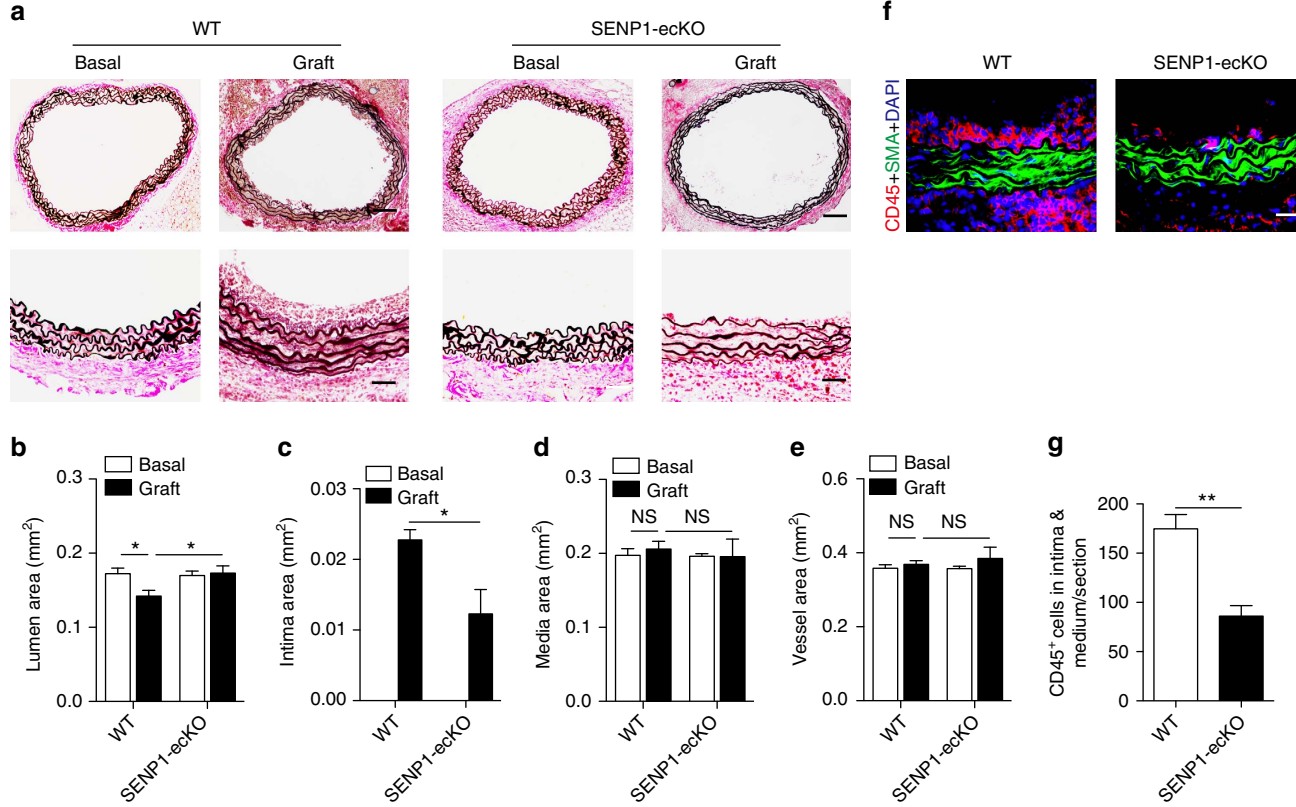

**Figure 2 | Endothelial SENP1 deficiency attenuates pathological remodelling in a mouse GA model.** A male thoracic aorta segment from WT or SENP1-ecKO mice was transplanted into the abdominal aorta of female C57BL/6 mice to establish the GA model or transplanted into male C57BL/6 mice as controls. Grafts were isolated at 2 weeks post-transplantation. (**a**) Histological analysis of grafts by EVG staining. Representative low-powered images (bar represents 200 μm) and high-powered images (bar represents 50 μm) are shown in the top row and the bottom row, respectively. (**b–e**) Morphometric assessment of artery lumen area, intimal area, media area and vessel area of each artery graft. (**f,g**) Endothelial knockout of SENP1 reduces inflammatory cell infiltration in grafts. Immunofluorescence analysis of artery grafts with anti-α-SMA and anti-CD45 antibodies was performed, followed by counterstaining with DAPI. Representative images are shown in (**f**) with quantification data in (**g**). Bar represents 50 μm. Data in (**b–e,g**) are presented as the mean ± s.e.m. from at least six mice per group. *P < 0.05 and **P < 0.01; unpaired t-test. EVG, Elastin-Van Gieson; NS, not significant.

Specifically, the decreased leukocyte accumulation within the vessel wall caused by the endothelial SENP1 deficiency suggests a function for SENP1 in regulating endothelial activation and leukocyte recruitment.

**SENP1 facilitates endothelial activation and dysfunction in GA.** To further determine the cellular function of SENP1 in GA, especially EC activation, we examined the expression levels of endothelial adhesion molecules early after transplantation, prior to discernable neointima formation. At 3 days post-operation, decreased mRNA expression of the inducible ICAM-1 and VCAM-1 was detected in SENP1-ecKO aortas compared with those in the WT group (Supplementary Fig. 4A,B). Immunofluorescence staining confirmed the downregulation of these adhesion molecules in the graft endothelium that lack SENP1 (Fig. 3a). To further identify inflammation-induced EC activation, primary MAECs were isolated from WT and SENP1-ecKO mice and subsequently treated with tumor necrosis factor (TNF) or interleukin-1β (IL-1β), the representative pro-inflammatory cytokines produced in GA. Consistent with the tissue assessment, MAECs from SENP1-ecKO mice exhibited less induction of the surface expression of endothelial adhesion molecules than WT cells following cytokine stimulation (Fig. 3b with quantification in Fig. 3c–e). In addition, the function of endothelial SENP1 was investigated in human vascular ECs with a catalytic inactive form of SENP1 (SENP1-Mut). In response to pro-inflammatory

cytokines, primary human umbilical vein endothelial cells (HUVECs) overexpressing SENP1-Mut exhibited a retarded induction of adhesion molecules at the mRNA level (Supplementary Fig. 5A–C) and adhesion molecule surface expression (Fig. 3f with quantification in Fig. 3g,h), indicating a regulatory function for SENP1 in EC activation.

Meanwhile, EC activation and dysfunction have been implicated in GA progression. To this end, we investigated whether endothelial SENP1 affects vasomotor function. The contraction of the graft in response to phenylephrine (PE) and the relaxation of the graft in response to acetylcholine (ACh) were examined by myography. Compared with WT graft aortas, SENP1-ecKO grafts showed reduced constriction in response to PE but increased relaxation in response to ACh (Supplementary Fig. 6A,B). To determine the effect of endothelial SENP1 deficiency on endothelial nitric oxide synthase (eNOS) activity in grafts, graft aortic rings were incubated with the NOS inhibitor L-nitro arginine methyl ester (L-NAME) to inhibit the basal release of eNOS-derived NO prior to contraction with PE. L-NAME treatment resulted in an increase in isometric tension, which reflected the absence of basal NO. However, the vasoconstrictive response to PE was similar in both WT and SENP1-ecKO grafts after L-NAME treatment (Supplementary Fig. 6C). These effects on vasomotion were specifically derived from the endothelium because the vasoconstrictive responses to KCl and relaxation responses to the NO donor

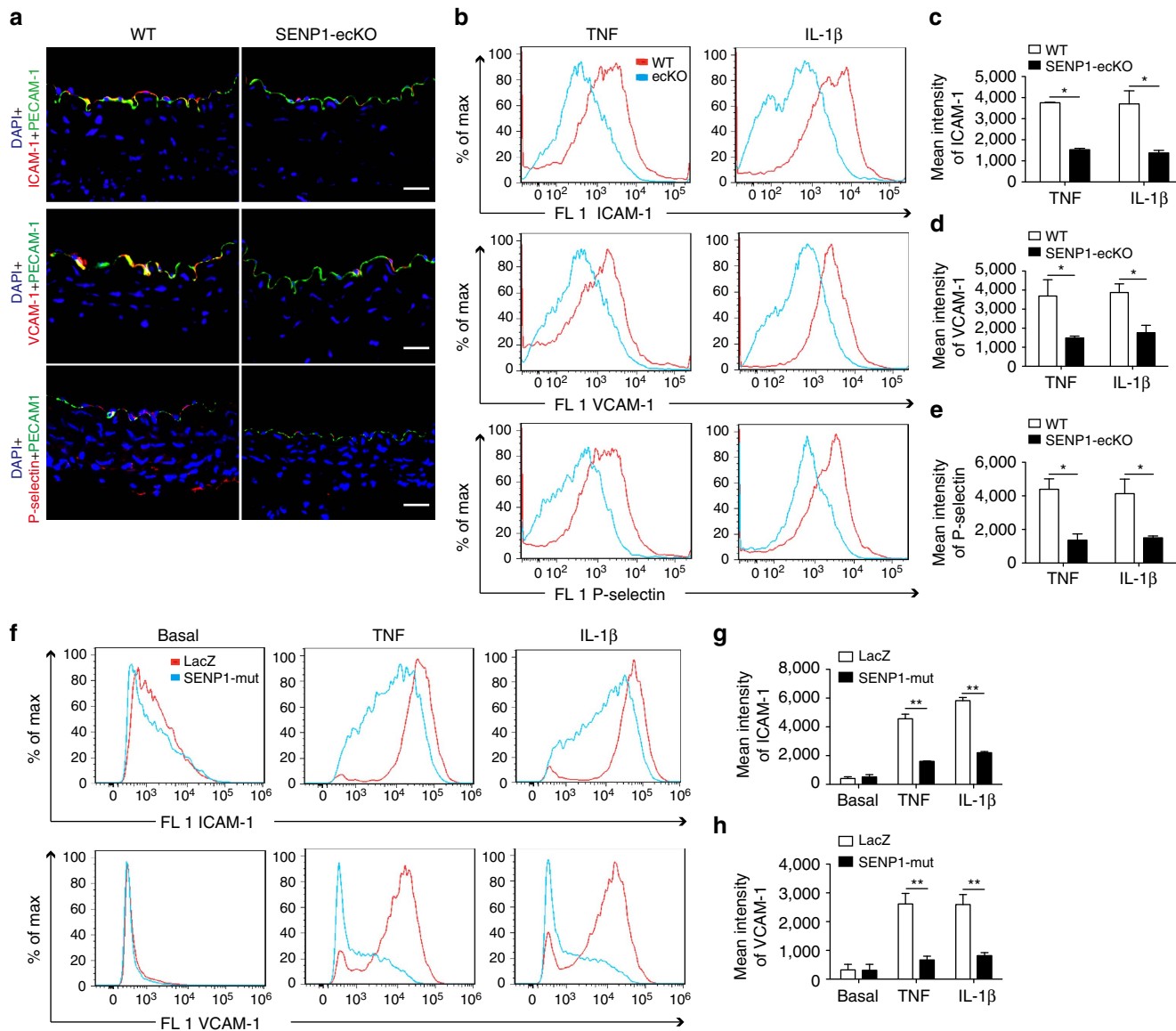

**Figure 3 | Loss of endothelial SENP1 inhibits EC activation.** (**a**) Grafts from WT or SENP1-ecKO mice were harvested 3 days post-transplantation. The induction of endothelial adhesion molecules was demonstrated by immunofluorescence staining of ICAM-1, VCAM-1, or P-selectin and PECAM-1 with DAPI labelling of the nuclei. Bar represents 50 μm. (**b–e**) Attenuated induction of adhesion molecules in SENP1-ecKO MAECs. Flow cytometry analysis of ICAM-1, VCAM-1 and P-selectin in MAECs isolated from WT or SENP1-ecKO mice after TNF or IL-1β treatment. Representative histograms are shown in (**b**) with the quantification of mean intensity in (**c–e**). (**f–h**) Overexpression of the catalytically inactive form of SENP1 (SENP1-Mut) inhibits the induction of adhesion molecules in HUVECs. HUVECs were infected by Ad-SENP1-Mut or vector control (Ad-LacZ) for 24 h, treated with pro-inflammatory cytokines and analysed by flow cytometry in the same way as MAECs. Representative histograms of ICAM-1 and VCAM-1 are shown in (**f**) with the quantification of mean intensity in (**g,h**). Data are presented as the mean ± s.e.m. from at least three independent experiments. *P < 0.05 and **P < 0.01; two-way ANOVA followed by Bonferroni post-test. MAEC, mouse aortic endothelial cell.

drug sodium nitro prusside were similar between the two groups (Supplementary Fig. 6D,E). Thus, endothelial SENP1 deficiency improves vasoreactivity in grafts by preserving eNOS activity and endothelial function. Interestingly, the mesenchymal markers α-SMA and Notch3 were found to be upregulated in the lumen ECs of WT grafts at a middle stage of GA (2 weeks post-transplantation), whereas attenuated α-SMA and Notch3 induction was observed in the luminal endothelium of SENP1-ecKO grafts (Supplementary Fig. 7A with quantification in Supplementary Fig. 7B,C). These results suggest that endothelial-to-mesenchymal transition (Endo-MT), another prototypic outcome of endothelial dysfunction[37], might be regulated by SENP1. Mesenchymal marker induction was not

detected at the early stage (3 days post-transplantation) in grafts from both groups (Supplementary Fig. 7D).

Therefore, SENP1 deficiency preserves the endothelial function of grafts and prevents further endothelial inflammation during GA progression. These results are in agreement with our clinical observations and support a role for SENP1 in driving GA pathogenesis by targeting EC function. Taken together, these data support a role for SENP1 in preventing GA pathogenesis, in which SENP1-regulated EC activation and consequent EC dysfunction may be a potential mechanism.

**SENP1 deficiency inhibits leukocyte–endothelial trafficking.** EC activation precedes the recruitment of various leukocytes and is,

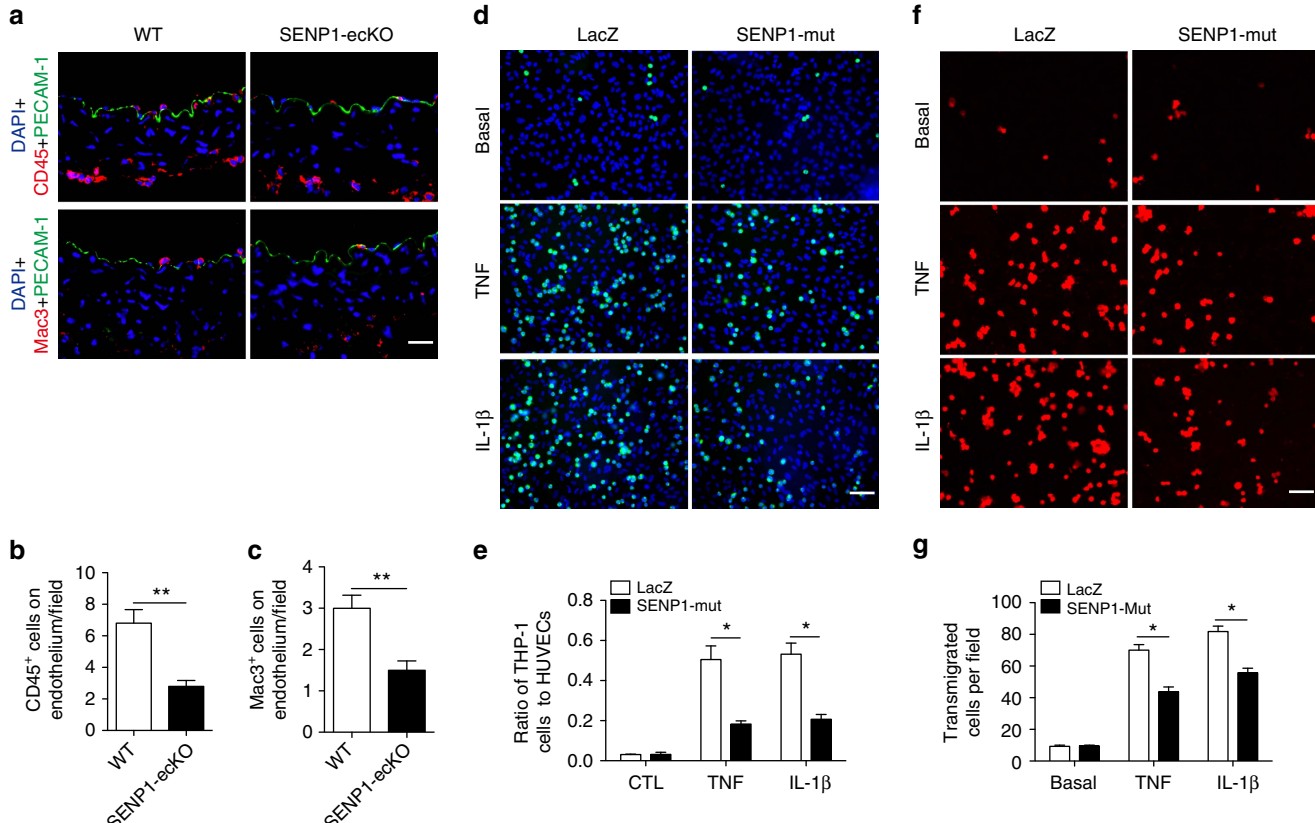

**Figure 4 | Endothelial SENP1 deficiency inhibits pro-inflammatory cytokine-dependent leukocyte infiltration across the endothelium.** Cross-sections of grafts from WT or SENP1-ecKO mice were analysed by immunofluorescence staining at 3 days after transplantation. Leukocyte infiltration was determined by immunostaining with the pan-leukocyte marker CD45 or the macrophage marker Mac3; PECAM-1 antibodies label the endothelium. The nuclei were stained with DAPI. Representative images are shown in (**a**), and the quantification of adherent cells is shown in (**b,c**). Bar represents 50 μm. Data are presented as the mean ± s.e.m. from five independent grafts. *$P < 0.05$ and **$P < 0.01$; unpaired $t$-test. (**d,e**) Overexpression of SENP1-Mut limits leukocyte–endothelial adhesion. Confluent monolayers of HUVECs were treated with TNF or IL-1β for 24 h after overexpressing SENP1-Mut or LacZ control by adenoviral infection. Afterwards, calcein green-labelled THP-1 monocytes were loaded onto the HUVEC layer for 1 h. Non-adherent cells were then washed away, and adherent cells were visualized using fluorescence microscopy. Representative images are shown in (**d**), and the normalized ratio of adherent THP-1 cells in each field is shown in (**e**). Bar represents 200 μm. Data are presented as the mean ± s.e.m. from five images in three independent experiments. *$P < 0.05$ relative to LacZ control; two-way ANOVA followed by Bonferroni post-test. (**f,g**) SENP1-Mut overexpression limits leukocyte–endothelial transmigration. SENP1-Mut- or LacZ-overexpressing HUVECs were cultured on the upper chamber of Transwell inserts that were coated with 0.1% gelatin and then treated with TNF or IL-1β for 24 h. Subsequently, calcein red-labelled THP-1 monocytes were seeded onto the activated HUVEC monolayers for 24 h. Transendothelial migrated monocytes were determined by visualizing cells in the lower compartment of the insert using fluorescence microscopy. Representative images are shown in **f** with normalized quantification of migrated THP-1 cells in **g**. Bar represents 50 μm. Data are presented as the mean ± s.e.m. from five images in three independent experiments. *$P < 0.05$; two-way ANOVA followed by Bonferroni post-test. HUVEC, human umbilical vein endothelial cell.

accordingly, the fundamental event of vascular inflammation during GA. To illustrate the biological function of endothelial SENP1 in GA, we examined inflammatory cells in grafts at 1 week after transplantation, which is the time window for observing initial leukocyte infiltration across the endothelium. Corresponding to endothelial adhesion molecule expression, decreased numbers of infiltrating acute inflammatory cells were observed in SENP1-ecKO grafts compared with WT grafts, as evidenced by CD45[+] and Mac3[+] cells on the EC layer (Fig. 4a with quantification in Fig. 4b,c). The major steps involved in inflammatory cell recruitment include leukocyte–endothelial adhesion and transmigration. To determine the role of endothelial SENP1 in this process, a pro-inflammatory cytokine-mediated human monocyte-vascular EC system was employed. Upon TNF or IL-1β pretreatment, the SENP1-Mut-overexpressing HUVEC monolayer exhibited a significantly reduced number of adhered and transmigrated monocytes (THP-1 cells), whereas the LacZ group had abundant

numbers of adhered and transmigrated monocytes (Fig. 4d–g). The results of both *in vivo* and *in vitro* cellular assays provide clues with which to explore the molecular mechanism of SENP1 in GA progression.

**Exaggerated GATA2 SUMOylation in SENP1-ecKO grafts.** SENP1 function requires deSUMOylase activity as the catalytically inactive form of SENP1 (SENP1-Mut) attenuated inflammation-induced EC activation. Concomitantly, clinical GA specimens indicated that SENP1 might be associated with GATA2, the transcription factor governing EC activation. In addition, knockdown of endothelial GATA2 in donor aorta by lentiviral GATA2 shRNA (Supplementary Fig. 8A) significantly reduced EC activation (Supplementary Fig. 8B), neointima formation (Supplementary Fig. 8C–E) and inflammatory cell infiltration in transplanted grafts (Supplementary Fig. 8F,G), indicating a direct link between endothelial GATA2 and GA

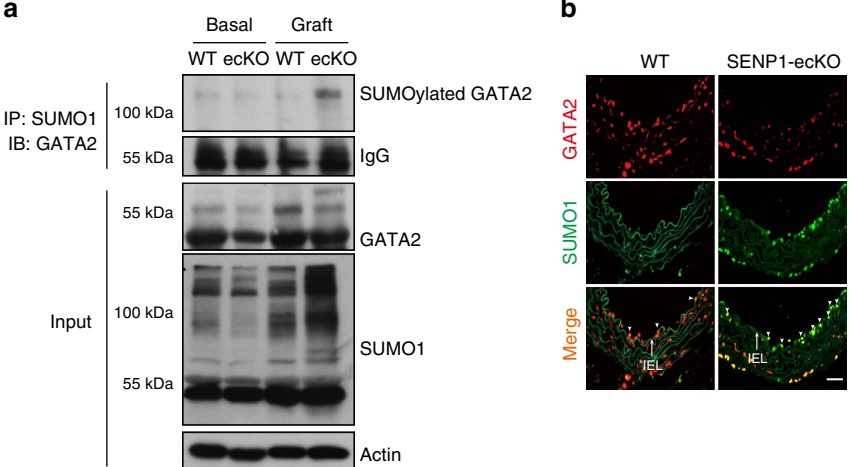

**Figure 5 | Augmented GATA2 SUMOylation in SENP1-ecKO grafts.** (**a**) Enhanced GATA2 SUMOylation in SENP1-ecKO grafts after transplantation for 3 days. Graft lysate was subjected to IP with anti-SUMO1 followed by western blotting with anti-GATA2 antibody. (**b**) Co-localization of SUMO1 and GATA2 on the endothelium of the graft artery was determined by immunofluorescence staining of SUMO1 and GATA2. Data are pooled from two independent studies. Representative images are shown. Arrows mark the internal elastic lamina to delineate the luminal endothelium from the media. The SUMO-associated GATA2 on the luminal endothelium is indicated by arrowheads. Bar represents 50 μm.

development. Therefore, we reasoned that the loss of SENP1 leads to GATA2 SUMOylation, hence reducing EC activation in GA. To test our hypothesis, we first examined whether enhanced levels of SUMOylated GATA2 could be detected in SENP1-ecKO grafts. As expected, SUMOylated GATA2 was remarkably increased in SENP1-ecKO grafts, which was confirmed by immunoprecipitation (IP) and immunofluorescence (IF) staining of GATA2 and SUMO1 (Fig. 5a,b). Along with the increased co-localization of GATA2 and SUMO1, attenuated GATA2 intensity was detected in the endothelium of SENP1-ecKO grafts, indicating that GATA2 SUMOylation may reduce its stability. To further determine the patho-physiological role of GATA2 SUMOylation in GA, we performed IF staining of GATA2 and SUMO1 on WT non-transplanted donor vessels (basal) and grafts harvested 3 days after transplantation. Decreased SUMOylated GATA2 in the endothelial layer was detected in grafts compared with the basal group (Supplementary Fig. 9), which validates that GATA2 deSUMOylation occurs naturally during GA. Enhanced endothelial GATA2 expression in grafts also suggests the negative effect of GATA2 SUMOylation on GATA2 stability.

**SENP1 directly regulates GATA2 SUMOylation in ECs.** To confirm the SUMO modification on GATA2, an *in vitro* SUMOylation assay was performed on recombinant GATA2 in the presence of SUMO E1, SUMO E2 and ATP. SUMO1 conjugation was identified as the dominant modification, although both SUMO-1 and SUMO-2 can be utilized to generate SUMOylated GATA2 (Supplementary Fig. 10). More importantly, endogenous GATA2 SUMOylation was detected in HUVECs by IP (Fig. 6a). To better understand the biochemical characteristics of GATA2 SUMOylation, SUMO-binding sites on GATA2 were inspected via bioinformatic analysis. Lysine 222 and 389, which are evolutionarily conserved among vertebrates (Supplementary Fig. 11), are identified as critical SUMOylation sites in GATA2; the IP assay demonstrated that the GATA2 K222R or K389R mutation led to diminished levels of SUMOylated GATA and the K222/389R (GATA2-2KR) mutation led to undetectable levels of SUMOylated GATA2 when co-expressed with SUMO1 (Fig. 6b). To determine whether SENP1 directly deSUMOylates GATA2, the IP assay was performed in isolated MAECs from either WT or SENP1-ecKO

mice. SENP1 deficiency significantly enhanced SUMOylated GATA2 levels, whereas normal levels of SUMOylated GATA2 were detected in WT cells (Fig. 6c). In addition to an endogenous interaction, GATA2 WT was co-expressed with SENP1-WT or SENP1-Mut. GATA2 SUMOylation was markedly diminished by SENP1-WT but enhanced by SENP1-Mut (Fig. 6d), suggesting that SENP1 directly regulates GATA2 SUMOylation. To explore the connection between SENP1-mediated GATA2 deSUMOylation and endothelial inflammation, HUVECs were treated with pro-inflammatory cytokines prior to the detection of SENP1 and SUMOylated GATA2. Increased SENP1 expression but decreased GATA2 SUMOylation was observed in the presence of TNF and IL-1β (Fig. 6e,f). Together with the enhanced SUMOylated GATA2 levels observed in the endothelium of SENP1-ecKO grafts, these results indicate an important role for endothelial SENP1 in vascular inflammation by regulating GATA2 SUMOylation.

**SUMOylation destabilizes and deactivates GATA2.** Reduced expression of adhesion molecules in SENP1-deficient ECs could be a result of alterations in the cellular localization, protein stability and/or protein–DNA binding activity of SUMOylated GATA2, as demonstrated by other transcription factors. To determine whether SUMOylation affects these protein characteristics of GATA2, we employed GATA2-WT, GATA2-2KR and SUMO1 fused to GATA2 (SUMO-GATA2, Supplementary Fig. 12A) constructs that were generated to mimic SUMOylated GATA2 for further investigation. Immunofluorescence staining showed a similar pattern of nuclear localization for GATA2-WT, GATA2-2KR and SUMO-GATA2 (Supplementary Fig. 12B). In terms of GATA2 stability, the half-lives of GATA2-WT, GATA2-2KR and SUMO-GATA2 were determined in the presence of the protein synthesis inhibitor cycloheximide (CHX). Interestingly, SUMO-GATA2 showed a lower expression level and decreased half-life compared with GATA2-WT, whereas GATA2-2KR, the SUMO-defective mutant, exhibited enhanced stability (Fig. 7a–d). The effect of GATA2 SUMOylation on its expression was further confirmed by western blotting after co-expressing GATA2-WT with SUMO1, SENP1-WT or SENP1-Mut. As expected, reduced GATA2 expression was observed when co-expressed with SUMO1. However, this effect was

augmented by SENP1-Mut and reversed by SENP1-WT co-expression (Fig. 7e,f). Additionally, the expression level of endogenous GATA2 was obviously decreased in SENP1-ecKO MAECs compared with WT cells (Fig. 6c, input). These results reveal that SUMOylation attenuates GATA2 stability via the functionally regulated SENP1, which confirms the

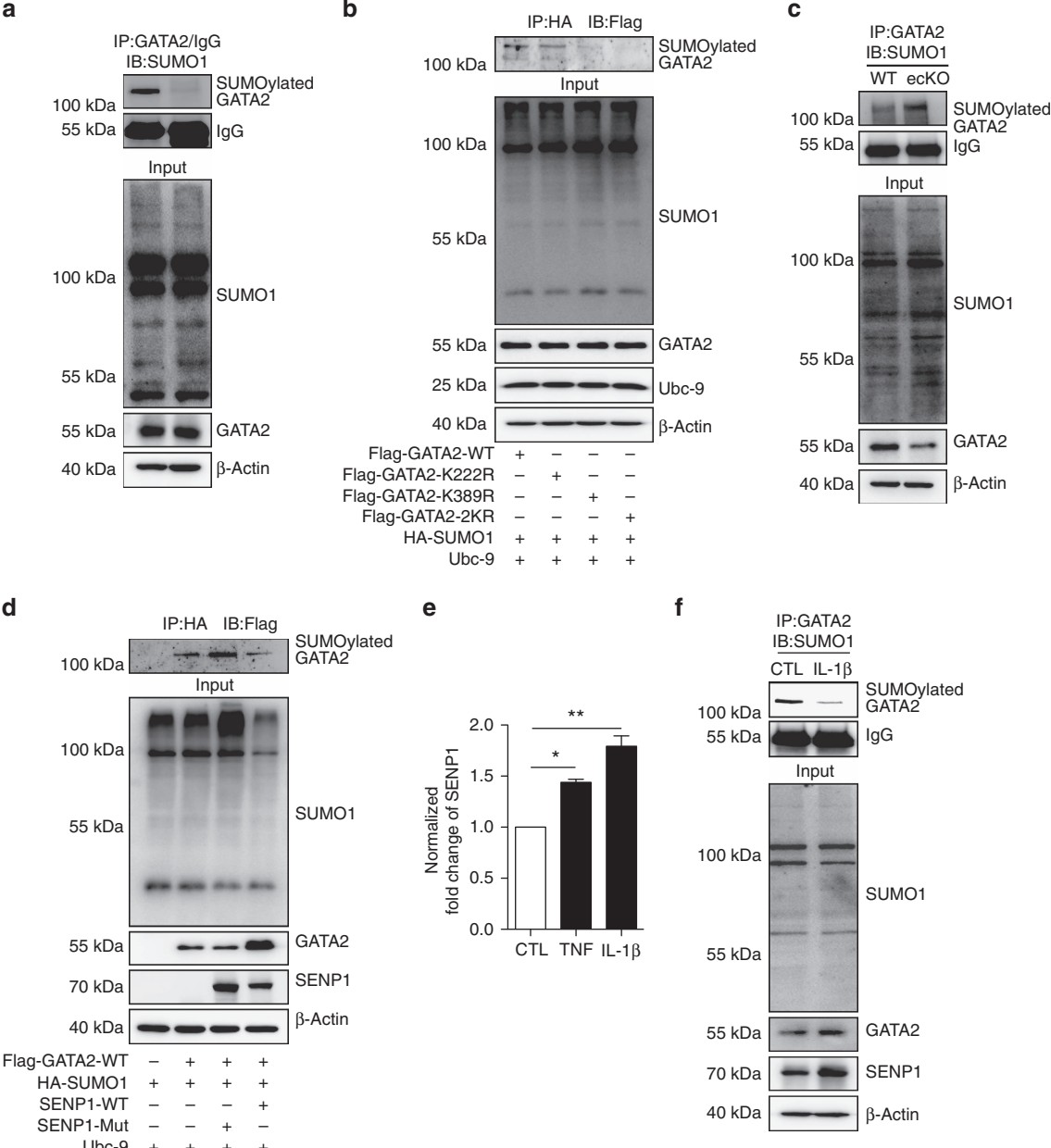

**Figure 6 | SENP1 regulates GATA2 SUMOylation. (a)** Endogenous GATA2 SUMOylation in HUVECs. HUVEC lysates with 20 mM NEM (deSUMOylase inhibitor) were subjected to IP with anti-GATA2 antibody or anti-IgG antibody followed by western blotting with anti-SUMO1 antibody. **(b)** Identification of the GATA2 SUMOylation sites. GATA2-WT, GATA2-K222R, GATA2-K389 R or GATA2-2KR (GATA2-K222/389R) was co-expressed with SUMO1 and Ubc-9 in 293T cells as indicated. In the presence of 20 mM NEM, cell lysates were harvested and subjected to IP with anti-HA antibody (SUMO1) followed by western blotting with anti-Flag antibody (GATA2). SUMOylated GATA2 is indicated. Input levels of SUMO1, GATA2 and Ubc-9 were detected by western blotting with anti-HA, anti-Flag and anti-Ubc-9, respectively. **(c,d)** SENP1 deSUMOylates GATA2. **(c)** Cell lysates of primary MAECs from WT or SENP1-ecKO mice were harvested in the presence of 20 mM NEM and subjected to IP with anti-GATA2 antibody. SUMOylated GATA2 was determined by western blotting with anti-SUMO1 antibody. Input levels of SUMO1 and GATA2 were detected by the anti-SUMO1 and anti-GATA2 antibodies, respectively. **(d)** GATA2-WT, SUMO1 and Ubc-9 were co-expressed with SENP1 or SENP1-CA in 293 T cells as indicated. Cell lysates with 20 mM NEM were subjected to IP with anti-HA antibody (SUMO1) followed by western blotting with anti-Flag antibody (GATA2). SUMOylated GATA2 is indicated. Input levels of SUMO1, GATA2 and SENP1 were detected by western blotting with anti-HA, anti-Flag and anti-Myc antibodies, respectively. **(e,f)** Increased SENP1 expression is concomitant with decreased GATA2 SUMOylation upon pro-inflammatory stimuli. HUVECs were treated with TNF or IL-1β for 24 h. Expression of SENP1 mRNA was determined by quantitative real-time PCR. Normalized fold change is shown in **(e)**. Data are shown as the mean ± s.e.m. from three independent experiments. *$P < 0.05$ and **$P < 0.01$; one-way ANOVA followed by Bonferroni test. **(f)** HUVEC protein lysates were harvested in the presence of 20 mM NEM. SUMOylated GATA2 was determined by IP with anti-GATA2 antibody followed by western blotting with anti-SUMO1 antibody. Input levels of SUMO1, GATA2 and SENP1 were indicated by anti-SUMO1, anti-GATA2 and SENP1 antibodies. NEM, N-ethylmaleimide.

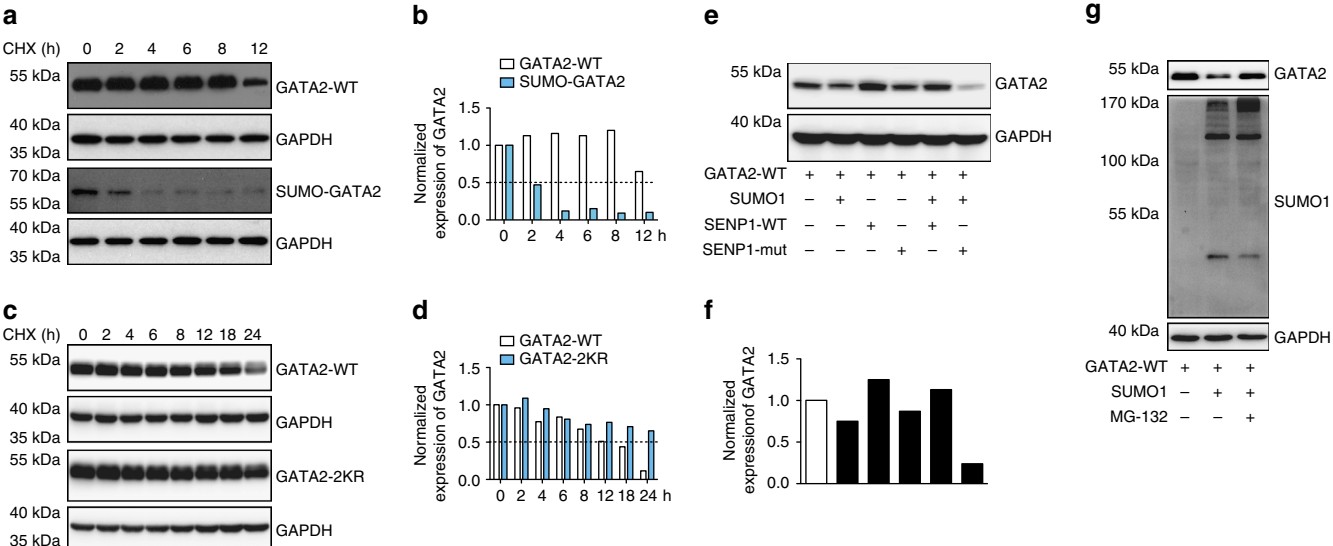

**Figure 7 | SENP1-mediated SUMOylation reduced GATA2 stability.** (**a**,**b**) SUMO conjugation reduces GATA2 half-life. GATA2 or SUMO-GATA2 was transfected into 293 T cells followed by treatment with 10 µg ml$^{-1}$ cycloheximide (CHX) for the indicated amount of time. Expression of GATA2 or SUMO-GATA2 was determined by western blotting with anti-Flag antibody. GAPDH was used as a control. Representative blots are shown in (**a**), and the quantitation of relative protein levels is shown in (**b**). (**c**,**d**) GATA2 SUMOylation site mutations elongate the half-life of GATA2. GATA2 or the GATA2-2KR mutant was transfected into 293 T cells followed by treatment with CHX for the indicated amount of time. Expression of GATA2 or GATA2-2KR was determined by western blotting with anti-Flag antibody. GAPDH was used as a control. Representative blots are shown in (**c**), and the quantitation of relative protein levels is shown in (**d**). (**e**,**f**) SENP1-mediated SUMOylation regulates GATA2 stability. GATA2 was co-transfected with SUMO1, SENP1-WT or SENP1-Mut in 293 T cells. Expression of GATA2 was detected by western blotting with anti-Flag antibody. Representative blots are shown in **e** with normalized quantitative intensities of GATA2 in (**f**). (**g**) MG-132, a pan proteasome inhibitor, rescues the diminished GATA2 expression induced by SUMO1. GATA2 was co-transfected with SUMO1 in 293 T cells followed by MG-132 or vehicle treatment as indicated. Expression of GATA2 and SUMO1 was determined by western blotting with anti-Flag and anti-HA antibodies. All the data in this figure were pooled from three independent experiments.

faint expression of endothelial GATA2 when co-localized with SUMO1 in mouse grafts. In addition, inhibited GATA2 expression induced by SUMO modification was rescued when MG-132, a pan proteasome inhibitor, was administered (Fig. 7g), indicating that GATA2 protein stability is associated with proteasomal degradation.

DNA binding capability is the signature property of a transcription factor. Therefore, we investigated GATA2-DNA binding activity by the EMSA assay using a GATA2 probe. SUMO-GATA2 markedly diminished the DNA binding activity of GATA2 compared with GATA2-WT, whereas GATA2-2KR exhibited stronger DNA binding activity than WT GATA2 (Fig. 8a), indicating that SUMOylation resulted in attenuated GATA2-DNA binding activity. By contrast, recruitment of GATA2 to promoters of adhesion molecule genes is critical for their expression and EC activation. To directly determine the effect of SENP1-mediated GATA2 SUMOylation on this process, we measured GATA2 binding to the promoter regions of adhesion molecules in HUVECs by a chromatin immunoprecipitation (ChIP) assay. Expression of SENP1 mutant protein significantly blunted the association of GATA2 with the promoters of ICAM-1, VCAM-1 and E-selectin in response to TNF (Fig. 8b, 11.36-fold increase vs 3.99-fold increase, 63.86 vs 6.39 and 7.70 vs 1.55, respectively, LacZ cells vs SENP1-mutant cells), which correlated with the inhibited EC activation and GA that resulted from endothelial SENP1 deficiency. Furthermore, adenoviruses encoding GATA2-WT, GATA2-2KR, SUMO-GATA2 or GFP were used to infect HUVECs in which endogenous GATA2 was first depleted by siRNA transfection. The ChIP assay showed that, consistent with the DNA binding activity, GATA2-WT recovered and GATA2-2KR enhanced adhesion molecule promoter binding when SUMO-GATA2 failed to be recruited (Fig. 8c).

All together, these data strongly support a critical role for SENP1 and SUMOylation in controlling GATA2 stability and recruitment to adhesion molecule promoters.

**Critical role of GATA2 SUMOylation in endothelial activation.** Finally, we determined the functional significance of GATA2 SUMOylation in endothelial activation. To this end, HUVECs were infected with adenoviruses expressing GATA2-WT, GATA2-2KR, SUMO-GATA2 or GFP and then treated with TNF. Again, GATA2 was knockdown by siRNA before adenoviral infection to eliminate the effect of endogenous GATA2. The expression of GATA2 and adhesion molecules was first examined by western blotting after 24 h of TNF treatment. As expected, TNF induced adhesion molecule expression in HUVECs infected with control siRNA, whereas a comparable expression pattern was observed in cells expressing GATA2-WT (Fig. 9a). Adhesion molecule induction was mildly strengthened but substantially reduced in the presence of GATA2-2KR and SUMO-GATA2, respectively (Fig. 9a). In addition, significantly increased expression of adhesion molecules in the early phase of TNF stimulation was observed in GATA2-2KR-expressing HUVECs compared with GATA2-WT-expressing cells by quantitative real-time PCR (Fig. 9b). These results are consistent with the effect of GATA2 SUMOylation on its transcriptional activity.

We further examined the function of GATA2 SUMOylation in leukocyte adhesion and transmigration. Corresponding to the trend of adhesion molecule expression described above, GATA2-2KR facilitated but SUMO-GATA2 diminished leukocyte adhesion and transmigration across ECs (Fig. 9c–f). These results illustrated the crucial function of GATA2 SUMOylation in EC activation and correlated pathological effects.

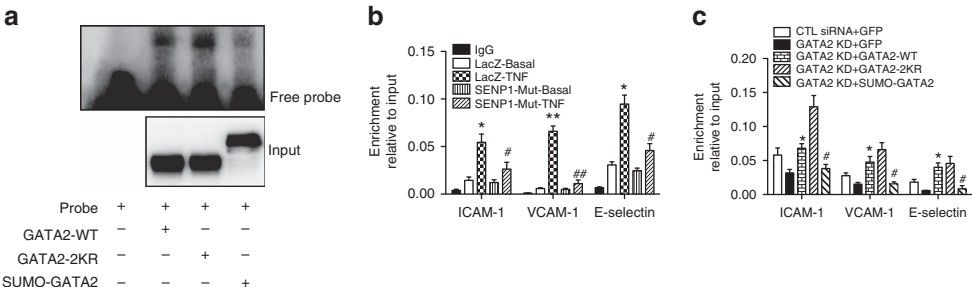

**Figure 8 | GATA2 SUMOylation inhibits its DNA binding activity.** (**a**) SUMO conjugation reduces GATA2 DNA binding activity. GATA2-WT, GATA2-2KR, SUMO-GATA2 or control constructs were transfected into 293 T cells, and nuclear extracts were processed with EMSA using a GATA2-specific oligonucleotide probe (top). Input of GATA2 was detected by western blotting with anti-Flag antibody (bottom). (**b**) Overexpression of catalytic inactive form of SENP1 (SENP1-Mut) inhibits recruitment of GATA2 to the promoter of ICAM-1, VCAM-1 and E-selectin. HUVECs were infected by Ad-SENP1-Mut or Ad-LacZ and then treated with TNF. Nuclear extracts were then subjected to ChIP assay with the anti-GATA2 antibody followed by quantitative real-time PCR for the promoter sequences of ICAM-1, VCAM-1 and E-selectin containing a GATA2 binding site. Quantitative results are shown as the ratio of ChIP to input values. Data are presented as the mean ± s.e.m. from three independent experiments. *$P < 0.05$ and **$P < 0.01$ compared with LacZ without TNF treatment (LacZ basal). #$P < 0.05$ and ##$P < 0.01$ compared with LacZ with TNF treatment; two-way ANOVA followed by Bonferroni post-test. (**c**) GATA2 SUMOylation attenuates its binding activity to the promoter of ICAM-1, VCAM-1, and E-selectin. HUVECs were transfected with GATA2 or control siRNA for 48 h and then infected with various GATA2 viruses or their vector controls as indicated. Cells were treated with TNF, and the nuclear extracts were subjected to the ChIP assay as described. Quantitative results are present as the ratio of ChIP to input values. Data are shown as the mean ± s.e.m. from three independent experiments. *$P < 0.05$ compared with GATA2 knockdown. #$P < 0.05$ compared with GATA2 knockdown with restored GATA2-WT; two-way ANOVA followed by Bonferroni post-test.

## Discussion

GA results in extensive vascular obstruction, and it is therefore considered to be a root cause of graft failure after organ transplantation. In addition to human leukocyte antigen expression on donor ECs, other factors including alarmin release, ischaemia–reperfusion, perioperative injury and cytomegalovirus infection all preferentially target graft endothelium; therefore, an endothelial phenotypic switch is the first step of the host response[1,3]. EC activation/dysfunction is the key event for GA initiation. Nevertheless, its modulatory mechanism is still unclear, which hinders the development of appropriate therapeutic approaches. In the present study, we used clinical GA specimens to explore the potential involvement of SUMOylation in the pathogenesis of GA. Here, we demonstrated an induction of the deSUMOylase SENP1 within the luminal endothelial layer of rejected coronary artery grafts, which is accompanied by the development of inflammation-mediated vascular remodelling. To verify the clinical observations, endothelial-specific SENP1 conditional knockout mice were generated and subjected to mouse aorta transplantation across the H-Y-dependent minor histocompatibility antigen barrier. Indeed, the clinical observation was supported by remarkably attenuated leukocyte infiltration and neointima expansion as well as endothelial dysfunction in SENP1-ecKO murine allografts. More importantly, the ameliorated vascular remodelling in grafts resulted from the halted activation of SENP1-deficient endothelium at the early stage of GA. Additionally, SENP1 deficiency or inactive mutated SENP1 in ECs blunted the induction of adhesion molecule expression. This result mirrors the finding of augmented endogenous SENP1 and enhanced adhesion molecule expression in the endothelium of clinical GA specimens. Critically, all of the data above indicate a regulatory role for SENP1 in mediating SUMOylation during GA progression.

The concurrent induction of endothelial SENP1 and GATA2 in clinical GA specimens as well as concomitant adhesion molecules expression propelled us to identify the intrinsic involved in this process. Strikingly, enhanced GATA2 SUMOylation was detected in the SENP1-ecKO graft, especially in the endothelium. Further investigation both in mice and in human ECs confirmed the constitutive GATA2 SUMOylation and revealed that such modifications reduced GATA2 stability and DNA binding activity, which then diminished the downstream expression of adhesion molecules and EC activation. In addition, inflammation-induced SENP1 directly regulated GATA2 deSUMOylation in ECs and therefore promoted leukocyte–endothelial adhesion and transmigration. Intriguingly, previous reports and our recent study showed that nuclear factor-κB (NF-κB), another important transcription factor in regulating endothelial adhesion molecules, is also regulated by SENP1 through alternating IκBα SUMOylation and NEMO SUMOylation in adipose tissue in vitro[38,39]. However, the role of SENP1-mediated SUMOylation in endothelial NF-κB signalling has not yet been determined. In the present study, loss of SENP1 attenuates NF-κB activation (Supplementary Fig. 13A) but enhances IκBα SUMOylation instead of NEMO SUMOylation (Supplementary Fig. 13B,C) in ECs. It has been well established that IκBα SUMOylation inhibits IκBα degradation, thereby attenuating NF-κB activation[38]. Thus, an endothelial SENP1 deficiency inhibits inflammatory NF-κB activation by promoting IκBα SUMOylation. Meanwhile, it was reported that GATA2 acted synergistically with NF-κB in a transcriptional complex when triggering gene transcription[40] and coordinately regulated endothelial adhesion molecule expression with NF-κB[41]. Here, we provide a direct link between endothelial GATA2 and GA (Supplementary Fig. 8) as well as the induction of endothelial adhesion molecules in vitro (Fig. 9a,b), which demonstrates the important role of GATA2 in endothelial activation and inflammation that is similar to NF-κB. Taken together, we conclude that SENP1-mediated SUMOylation suppresses the activity of both NF-kB and GATA2 in ECs and, therefore, the synergistic effect of the two transcriptional factors on endothelial inflammation in GA (Fig. 10).

As a type of vascular remodelling, GA development is inevitably dependent on inflammation in the vessel wall, which is the result of the host pathological response to the graft. Donor vascular endothelium lines the interface between graft tissue and recipient blood; thus, it is at the frontline of the host immune system. Interestingly, ischaemic and perioperative injuries immediately break endothelial haemostasis in grafts by triggering the release of intracellular IL-1β and IL-6 stored in vascular ECs and resident macrophages, even before the release of pro-inflammatory cytokines by the host immune system. Upon

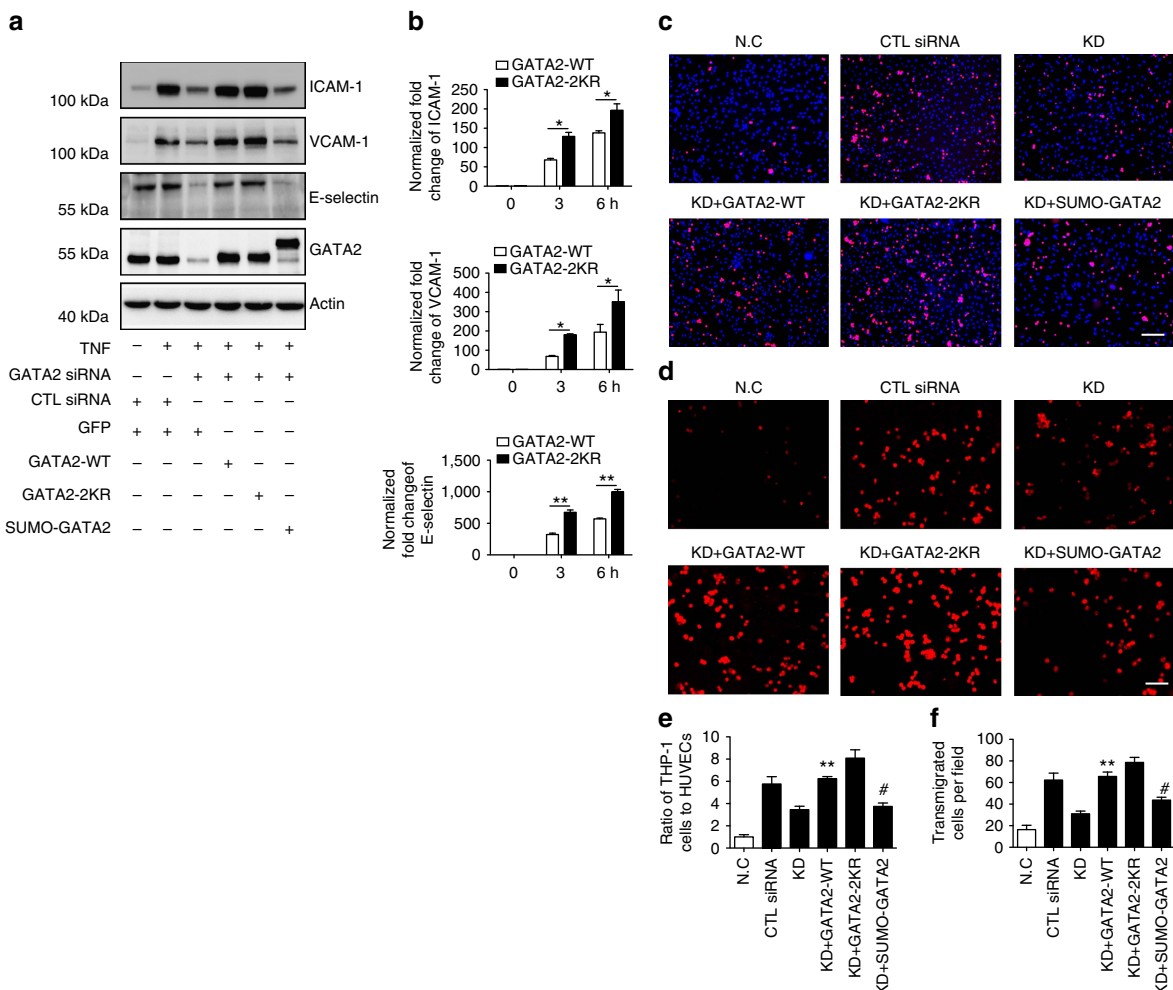

**Figure 9 | The critical role of GATA2 SUMOylation in EC activation.** (**a**) GATA2 SUMOylation regulates the expression of endothelial adhesion molecules. HUVECs were transfected by GATA2 siRNA or control siRNA for 48 h followed by infection with Ad-GATA2-WT, Ad-GATA2-KR, Ad-SUMO-GATA2 or their vector control (GFP) as indicated. HUVECs were treated with TNF or vehicle control for 24 h, and the cell lysates were then subjected to western blotting with anti-ICAM-1, anti-VCAM-1, anti-E-selectin or anti-GATA2 antibodies. Actin was used as a loading control. (**b**) GATA2 SUMOylation site mutation increases endothelial adhesion molecule induction by TNF at the early phase. The expression of endothelial adhesion molecules was determined by quantitative real-time PCR in HUVECs with GATA2-WT or GATA2-2KR reconstitution as described in (**a**). Data are presented as the mean ± s.e.m. from at least three independent experiments. *$P < 0.05$ and **$P < 0.01$; two-way ANOVA followed by Bonferroni post-test. (**c,e**) GATA2 SUMOylation inhibits leukocyte–endothelial adhesion. HUVECs were gene modified and treated as described in (**a**). Calcein green-labelled THP-1 monocytes were loaded onto a confluent HUVEC monolayer for 1 h. Non-adherent cells were then washed away, and the adherent cells were visualized using fluorescence microscopy. Representative images are shown in (**c**), and the normalized ratio of adherent THP-1 cells in each field is shown in (**e**). Bar represents 200 μm. Data are shown as the mean ± s.e.m. from three independent experiments. **$P < 0.01$ compared with GATA2 knockdown. #$P < 0.05$ compared with GATA2 KD with restored GATA2-WT; one-way ANOVA followed by Bonferroni test. (**d,f**) GATA2 SUMOylation inhibits leukocyte–endothelial transmigration. HUVECs were transfected with GATA2 siRNA or control siRNA for 48 h and then infected with various GATA2 viruses or their vector control as indicated. HUVECs with or without GATA2 restoration were cultured on the upper chamber of the Transwell insert that was coated with 0.1% gelatin and then treated with TNF or IL-1β cytokines for 24 h. Subsequently, calcein red-labelled THP-1 monocytes were seeded onto the activated HUVEC monolayers for 24 h. Transendothelial migrated monocytes were determined by visualizing cells in the lower compartment of the insert using fluorescence microscopy. Representative images are presented in (**d**) with normalized quantification of migratory THP-1 cells in (**f**). Bar represents 50 μm. Data are shown as the mean ± s.e.m. from three independent experiments. **$P < 0.01$ compared with GATA2 knockdown. #$P < 0.05$ compared with GATA2 knockdown with restored GATA2-WT; one-way ANOVA followed by Bonferroni test.

the nascent and successive inflammatory stimulations, ECs switch from a quiescent status to an activated state, with highly expressed adhesion molecules as signals for leukocyte recruitment. In the current study, we demonstrated the tight connection of post-translational SUMO modification to adhesion molecule induction throughout GA progression, thus defining the important role of SUMOylation in graft endothelial inflammation. Notably, SENP1, the major deSUMOylase, was identified as

a direct mediator in this event by coordinating the systemic modulation of gene expression in GA pathogenesis. To our knowledge, this study is the first to illustrate the role of SENP1 in cardiovascular diseases.

Recently, Heo *et al.*[34,42] reported the involvement of SENP2 in disturbed flow-mediated atherosclerosis, an arterial vascular remodelling disease with lipids as its critical initiating factor. Histologically, native atherosclerosis is usually detected in major

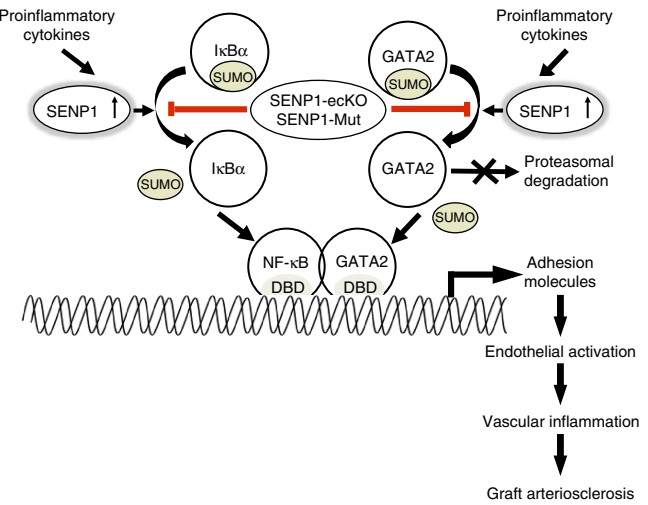

**Figure 10 | Model for the critical role of endothelial SENP1 in graft arteriosclerosis (GA).** Constitutive GATA2 SUMOylation leads to maintenance of its protein level and DNA binding in quiescent endothelial cells. Upon stimuli of pro-inflammatory cytokines during immune-mediated rejection, increased endothelial SENP1 promotes deSUMOylation of GATA2 and IκBα, which enhances GATA2 stability and GATA2-gene promoter binding capability as well as NF-κB activity. Consequently, the synergistic actions of GATA2 and NF-κB on the induction of endothelial adhesion molecule expression are suppressed. Endothelial SENP1 deficiency or dysfunction results in increased SUMOylation of GATA2 and IκBα, which blocks endothelial activation and subsequent vascular inflammation in the graft, thus preventing GA progression.

epicardial arteries and involves focal plaques with eccentric lesions containing a complex of deposited lipids, calcification and infiltrated cells[4]. GA appears in the entire vascular network of the graft and develops diffused lesions with neointima that is mainly composed of infiltrated leukocytes and smooth muscle cells. Accordingly, a common surrogate may be applicable for pathogenesis investigation and drug development in GA and in vascular remodelling diseases beyond atherosclerosis, such as restenosis following angioplasty, stenting or bypass grafting and even vein lesions. Specifically, endothelial inflammation plays a central role in GA, which is characterized by the maintenance of an intact endothelium with a subendothelial accumulation of infiltrated leukocytes. Meanwhile, atherosclerotic plaques exhibit endothelial erosion and smooth muscle cell accumulation in the intimal region adjacent to the media, suggesting the involvement of adventitia and media-mediated vascular inflammation. Therefore, our study could be a valuable addition to elucidating the mechanisms of endothelial inflammation in cardiovascular diseases.

The progression of endothelial inflammation begins with endothelial activation and the induction of adhesion molecule expression, which is prominently modulated at the transcriptional level. In addition to NF-κB, GATA2 is the other major transcription factor that controls the expression of endothelial adhesion molecules, including VCAM-1, P-selectin and E-selectin. Our study further demonstrates the direct involvement of endothelial GATA2 in GA development. GATA2 requires post-translational modifications, such as acetylation, ubiquitination and phosphorylation, to exert its functions[11,43–45]. Acetylation and ubiquitination of GATA2 have been shown to be involved in haematopoietic stem cell growth by regulating GATA2 DNA-binding activity and degradation, respectively[43,44]. In adipose tissue, GATA2 phosphorylation is critical for adipocyte maturation/inflammation[45]. However, the role of GATA2 in cardiovascular disease is still controversial[46–48], and the

pathophysiological role of its SUMOylation is largely unknown. Our previous study revealed that GATA1, another major member of the GATA family, is dynamically regulated by SUMO modification during developmental erythropoiesis[31]. However, minor changes of GATA2 SUMOylation and its expression were detected during mouse development. In the present study, the clinical assessment of GA showed that enhanced endothelial GATA2 expression is concurrent with adhesion molecule induction and, more intriguingly, with SENP1 expression, indicating the correlation of the SUMO system to endothelial GATA2 function in GA. In fact, our research group demonstrated that GATA2-driven EC activation was significantly attenuated in vascular ECs with SENP1 deficiency or dysfunction; SENP1 directly mediates GATA2 SUMOylation and thereby regulates both its stability and DNA binding capability during endothelial inflammation. Whether other SENP members could deSUMOylate GATA2 during GA remains to be determined. Of note, we showed that TNF-induced ICAM1 expression is also under the regulation of GATA2, which seems contradictory to a previous report[17]. It was reported that GATA2 could associate with NF-κB in a transcription factor complex and act synergistically with NF-κB in triggering gene transcription[40]. Therefore, upon TNF stimulation, GATA2 may not directly bind to the ICAM-1 promoter, but it can be recruited to the ICAM-1 promoter region by associating with NF-κB in the transcriptional complex. Collectively, our study uncovered a novel molecular mechanism for GA pathogenesis and illustrated the important role of GATA2 and its SUMOylation in vasculopathy.

In recent years, increasingly more attention has been paid to SUMOylation in vascular defects, especially in Dr Jun-ichi Abe's lab. They revealed that the SUMOylation of P53 or ERK5 plays specific roles in endothelial apoptosis and eNOS-related endothelial function in response to shear stress[33,34,49]. Combining our present findings about GATA2 and IκBα SUMOylation in GA, it is very likely that the modulation of the SUMO system in a selective transcription factor network within the endothelium is critical in the molecular pathogenesis of vasculopathy. These phenomena also correspond to the characteristics of SUMOylation in vitro, whereby the target molecule is found among multiple substrates under stress or in specific cell types, just as different regulatory patterns of SUMO modification on GATA1 and GATA2 are found in different physiological or pathological milieus. However, it is not known whether SUMOylation in vascular cells aside from ECs is important for the development of vascular diseases. If so, further investigations are warranted to determine how the SUMO system is regulated in these processes.

## Methods

**Clinical specimens.** Tissue collection and analyses was approved by the Yale Human Investigation Committee and the New England Organ Bank. Human coronary arteries were collected with informed consent from cardiac transplant recipients with chronic rejection or healthy organ donors. Arteries were procured in the operating room, and the disease was diagnosed macroscopically by an experienced cardiac surgeon and coded without patient identifiers. Vascular samples were embedded in optimal cutting temperature compound and immediately frozen[36].

**Mice.** Mice were kept in the animal facilities at Zhejiang University with a 12 h to 12 h light cycle and a humidity- and temperature-controlled environment. All animal experimental protocols were approved by the Institutional Animal Care and Use Committee of Zhejiang University. SENP1 vascular EC knockout mice (SENP1-ecKO) were generated by mating SENP1 flox (SENP1 flox/flox) mice that were created in our previous study[31] with vascular EC-specific VE-cadherin Cre transgenic mice (B6.Cg-Tg(Cdh5-cre)7Mlia/J; Jackson Laboratories, ME, USA). Both WT and SENP1-ecKO mice were backcrossed onto the C57BL/6 (B6) background, and they were viable and fertile.

**Aorta transplantation.** The procedure of mouse aortic allograft transplantation was performed as previously described[36]. Briefly, the thoracic aorta of 5-week-old WT male donor or SENP1-ecKO male donor mice was isolated, resected and

flushed with cold 100 U ml$^{-1}$ heparin saline solution. After proximal and distal clamping, the segment of thoracic aorta was surgically interposed into the abdominal aorta of female C57BL/6 (B6) recipients at 8–12 weeks of age using an end-to-end microsurgical anastomotic technique in an interrupted fashion. After removing the clamps, host mounted an alloimmune response against the male-specific H-Y minor histocompatibility antigen expressed by the graft, which induced endothelial activation by leukocyte-derived pro-inflammatory cytokines and resulted in intimal hyperplasia. Another group of B6 male donor mice were injected with lentivirus containing GATA2 shRNA or scrambled shRNA that is driven by the endothelial Tie2 promoter ($1.5 \times 10^8$ TU per mouse; Cyagen Biosciences, Guangzhou, China) via the jugular vein 1 week before transplantation.

**Histology and immunofluorescence.** To harvest the aortic grafts, the recipient mice were anaesthetized and perfused through the left ventricle at a pressure of 100 mm Hg with heparin saline solution for 5 min at room temperature. For haematoxylin–eosin staining and Elastic-Van Gieson staining, grafts were excised and fixed in IHC zinc fixative (BD Pharmingen) for 24 h at room temperature prior to embedding in paraffin. Sections of 5-µm thickness were made and processed according to a standard staining protocol. The perimeters of the endothelium and the internal and external elastic lamina were manually outlined, and the lumen area (within the endothelium), intimal area (between the endothelium and internal elastic lamina), media area (between the internal and external elastic lamina) and vessel area (within the external elastic lamina) were measured using image analysis software (ImageJ). Five sections from each graft were analysed, and the data were averaged for this graft. For immunofluorescence staining, grafts were isolated, embedded in optimal cutting temperature and cut into 5-µm thick sections. The slides were air-dried for 1 h at room temperature and then fixed in acetone for 10 min at 4 °C before staining. After washing in phosphate-buffered saline (PBS), sections were blocked in blocking buffer (10% goat serum and 1% bovine serum albumin in PBS) for 1 h at room temperature and incubated with primary antibody overnight at 4 °C. For GATA2 and SENP1 staining, slides were washed three times in PBS and permeabilized in PBS containing 0.1–0.3% Triton X-100 for 10–20 min before blocking. Antibodies for Rat anti-BrdU (Abcam, #ab6326, 1:200), Rat anti-CD45 (BD Biosciences, #550539, 1:200), FITC coagulated anti-α-SMA (Sigma Aldrich, #F3777, 1:400), Rabbit anti-CD62E (Abcam, #ab18981, 1:100), Rabbit anti-GATA2 (Abcam, #ab173817, 1:100), Goat anti-GATA2 (Santa Cruz, #sc16044, 1:50), Rat anti-ICAM-1 (Biolegend, #116102, 1:200), Rat anti-VCAM-1 (BD Biosciences, #553530, 1:200), Rat anti-PECAM-1 (BD Pharmingen, #553370, 1:200), Goat anti-PECAM-1 (Santa Cruz, #sc1506, 1:100), Goat anti-SENP1 (Santa Cruz, #sc46634, 1:50) and Goat anti-SUMO1 (Santa Cruz, #sc6376, 1:100) were diluted in blocking buffer. Goat IgG control (Abcam, #ab37373, same final concentration as SENP1 and GATA2) and blocking peptides (Santa Cruz, #sc-46634 P for SENP1, 100 × molar excess to SENP1; #sc-16044 P for GATA2, 100 × molar excess to GATA2) were used as negative controls. Fluorescence-labelled secondary antibodies, including Alexa Fluor 594-conjugated Donkey anti-Rabbit IgG (Jackson ImmunoResearch, #711-586-152, 2 µg ml$^{-1}$), Alexa Fluor 594-cojugated Donkey anti-Rat IgG (Jackson ImmunoResearch, #712-586-150, 2 µg ml$^{-1}$) and Alexa Fluor 488-cojugated Donkey anti-Goat IgG (Jackson ImmunoResearch, #705-545-147, 2 µg ml$^{-1}$), were incubated for 1 h at room temperature. 4,6-Diamidino-2-phenylindole (DAPI, 1 µg ml$^{-1}$) was used to stain the nucleus. All the sections were mounted and imaged using an inverted microscope (Zeiss, Axiovert 200M) under a × 40 objective and further analysed using Velocity software.

**Aortic ring assay.** The graft was dissected and cut into 3-mm long cylindrical segments. The rings were then mounted in a vessel myography system (Danish Myotechnologies, Aarhus, Denmark) and bathed in oxygenated Krebs buffer at a resting tension of 9.8 mN.[50,51] After equilibration, the concentration–response curves for PE with or without the NOS inhibitor L-NAME were generated to evaluate the vasoconstrictor responses. For vasodilatation, the rings were pre-constricted with a submaximal concentration of PE before the injection of ACh ($10^{-9}$–$10^{-4}$ M) or sodium nitro prusside ($10^{-9}$–$10^{-6}$ M) to generate response curves.

**Plasmids and adenovirus.** The expression plasmid for Flag-tagged GATA2-WT was obtained from Addgene (Addgene plasmid 1418). Expression plasmids for SUMO1, SENP1-WT and SENP1 CA mutant (SENP1-Mut) were previously described[31]. All mutations in GATA2 were generated by site-directed mutagenesis by mutating Lys 222 and 389 to Arg, which were verified by DNA sequencing. The SUMO-GATA2 fusion plasmid was constructed similarly to other SUMOylated proteins[31,52,53]. Briefly, the PCR-amplified SUMO1 fragment (without the double-glycine-encoding sequence at the C-terminal) was subcloned into the pXF6F vector plasmid using restriction sites BglII and NotI. Then, the PCR-amplified human GATA2 cDNA was subcloned into restriction sites XhoI and XbaI at the C-terminal domain of the SUMO1 fragment (Supplementary Fig. 12A). Adenoviral vectors expressing SENP1-Mut, GATA2-WT, GATA2-2KR and SUMO-GATA2 were constructed as we previously described[54].

**Cell culture.** Primary MAECs from WT or SENP1-ecKO mice were isolated and cultured as described[55]. HUVECs were obtained from the tissue culture core facility of the Yale Vascular Biology and Therapeutic (VBT) program and maintained in EBM-2 (Lonza) that was supplemented with 10% fetal bovine serum (Gibco), 2 mM L-glutamine, penicillin/streptomycin and the EGM-2 bullet kit in a humidified 37 °C incubator with 5% CO$_2$ atmosphere.

HUVECs were allowed to grow to 50% confluence before GATA2 siRNA transfection. Knockdown of GATA2 was performed with Lipofectamine RNA iMAX according to the manufacturer's instructions (Life Technologies). In brief, HUVECs were transfected with GATA2 siRNA (Life Technologies) or scrambled siRNA at 20 nM in reduced serum medium (Life Technologies). The reduced serum medium was replaced with fresh complete medium at 6 h after transfection, and the cells were kept in culture for 72 h before the experiment.

For GATA2 adenoviral infection, HUVECs were infected by various GATA2 adenoviruses with polybrene (5 nM) and incubated for 48 h before further experiments.

**Immunoprecipitation and immunoblotting.** After various treatments, the cells were washed twice with cold PBS and lysed in 1 ml of cold lysis buffer containing 50 mM Tris-HCl, pH 7.6, 150 mM NaCl, 0.1% Triton X-100, 0.75% Brij96, 1 mM sodium orthovanadate, 1 mM sodium fluoride, 1 mM sodium pyrophosphate, 10 mg ml$^{-1}$ aprotinin, 10 mg ml$^{-1}$ leupeptin, 2 mM PMSF and 1 mM EDTA for 20 min on ice[56]. For the detection of SUMOylated proteins by IP, 20 mM N-ethylmaleimide (deSUMOylase inhibitor) was added into the lysis buffer. Cell lysates were preincubated with antibody (1:100) on a rotator at 4 °C for 3 h prior to incubating with 20 µl of protein G (GE Healthcare) beads at 4 °C for 3 h. Immune complexes were collected by centrifugation at 2,000 r.p.m. for 2 min. After four washes, the samples were subjected to SDS–PAGE and blotted. The membranes were incubated with primary antibodies against SENP1 (Abcam, #ab108981, 1:1,000), GATA2 (Abcam, #ab109241, 1:1,000), SUMO1 (Cell Signaling Technology, #4930s, 1:1,000), GAPDH (Cell Signaling Technology, #5174, 1:1,000), β-actin (Cell Signaling Technology, #8457, 1:1,000), HA (Roche, #11 867 423 001, 1:500), Flag (Cell Signaling Technology, #8146, 1:1,000), HIF1α (Sangon, #D262108, 1:1,000), HIPK1 (Sangon, #261743, 1:1,000), p-p65 (Cell Signaling Technology, #3033s, 1:1,000), p65 (Cell Signaling Technology, #8242T, 1:1,000), ICAM-1 (Sangon, #D221861, 1:1,000), VCAM-1 (Novus, #NBP1-47491, 1:1,000), CD62E (Abcam, #ab18981, 1:1,000), NEMO (given by Dr Wang Ming, 1:1,000), or IκBα (Cell Signaling Technology, #4814T, 1:1,000) overnight at 4 °C and then incubated with horseradish peroxidase-linked anti-rabbit IgG secondary antibody (Jackson ImmunoResearch, #711-035-152, 1:10,000), anti-mouse IgG secondary antibody (Jackson ImmunoResearch, #715-035-151, 1:10,000) or anti-goat IgG secondary antibody (Jackson ImmunoResearch, #705-035-147, 1:10,000) for 1 h at room temperature. The blots were developed using a chemiluminescence ECL detection kit (Biological industries). The full-size images are presented in Supplementary Figs 14–17.

**Quantitative real-time PCR.** mRNA levels of pro-inflammatory and adhesion molecules in graft or HUVECs were quantified by quantitative real-time PCR. Tissue homogenates or cell lysates were generated using TRK lysis buffer (Omega). Total RNA was then isolated using the E.N.Z.A. total RNA Kit (Omega) according to the manufacturer's instructions. cDNAs were made using the ReverTra Ace qPCR RT kit (Toyobo). Quantitative real-time PCR was performed using the BioRad iCycler real-time PCR detection system (BioRad). GAPDH was used as an internal control. All of the primers are listed in Supplementary Table 1. The relative amount of mRNA was quantified and normalized to the control group.

**Flow cytometric analysis of specific cell surface markers.** To analyse the surface expression of adhesion molecules in ECs, the cells were stained with standard protocols using anti-mouse ICAM-1 PE (eBioscience, #85-12-0542-82, 1.25 µg ml$^{-1}$), anti-mouse VCAM-1FITC (eBioscience, #85-11-1061-81, 2.5 µg ml$^{-1}$), anti-mouse P-selectin PE (eBioscience, #85-12-0626-80, 2.5 µg ml$^{-1}$), anti-human ICAM-1 PE (eBioscience, #85-12-0549-41, 2.5 µg ml$^{-1}$) and anti-human VCAM-1 FITC (eBioscience, #85-11-1069-42, 5 µg ml − 1). Fluorescence-activated cell sorting analysis was performed using an LSRII flow cytometer (BD Biosciences) and the FlowJo software (Tree Star, Inc.). Isotype controls were used in each experiment.

**EMSA and ChIP assay.** The GATA2-specific oligonucleotide probe was obtained from Santa Cruz. Nuclear proteins were extracted using a nuclear protein and cytoplasm protein extraction kit (Beyotime, P0027). Forty femtomoles of labelled probe was incubated with 2 µg of extracted nuclear protein for 20 min at room temperature. Labelled biotin was analysed with a light shift chemiluminescent EMSA kit (Pierce, 20148) according to the manufacturer's instructions. The ChIP assay was performed as per the EZ ChIP Chromatin IP Kit (Millipore) manual's instructions and analysed by quantitative real-time PCR. All of the primers are listed in Supplementary Table 1. Antibodies included anti-GATA2 (Abcam), a positive control (anti-RNA polymerase II) and a negative control (normal IgG).

**Leukocyte–endothelial adhesion and transmigration assays.** For the leukocyte–endothelial adhesion assay, confluent HUVEC monolayers were pretreated with TNF or IL-1β after the overexpression of adenoviral SENP1-Mut, various GATA2 constructs or vector control. Calcein (Molecular Probes)-labelled THP-1 monocytes were co-cultured with stimulated HUVECs for 60 min, gently washed twice to remove non-adherent cells and microphotographs were acquired. At least five random fields were quantified per condition.

The transmigration assay was performed using 6.5 mm Transwell filters with 8-µm pores (Costar). HUVECs were seeded onto the insert, which was coated with 0.1% gelatin (Sigma-Aldrich), and cultured until confluent. HUVEC monolayers were stimulated with TNF or IL-1β for 24 h before calcein-labelled THP-1 was added into the chamber. After incubating for 24 h, the number of cells that transmigrated to the bottom compartment was imaged and quantified.

**Statistical analysis.** Statistical analyses were performed using GraphPad Prism 5 (GraphPad software). Data are represented as the mean ± s.e.m. Comparisons between two groups were made by unpaired $t$-test, and comparisons between more than two groups were made by one-way ANOVA followed by Bonferroni test or two-way ANOVA followed by Bonferroni post-test. $P$-values were two-tailed, and values $<0.05$ were considered to indicate statistical significance.

**Data availability.** The authors declare that all of the relevant data are available from the corresponding authors upon reasonable request.

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

## Acknowledgements

This work was supported by the National Natural Science Foundation of China (81422005, 81270357, 81600354, and 31470057), the Zhejiang Provincial Natural Science Foundation of China (LR14H020002), the Fundamental Research Funds for the Central Universities of China, and the China Postdoctoral Science Foundation (2015M581924). This work was partly supported by National Key Research and Development Program of China (2016YFC1300600), National Natural Science Foundation of China (No. 91539110) to WM, R01 HL109420 and R01 HL115148, and CT Stem Cell Innovation Award (Established Investigator Grant) 14-SCB-YALE-17 to WM.

## Author contributions

L.Y. supervised the overall project. L.Y. and C.Q. designed the research. C.Q., Y.W., HG.Z., L.Q., Y.S., X.Z., L.S., X.Z., J.C., H.Z. and H.F.Z. performed the experiments and analysed the data. G.T. and W.M. provided critical comments. C.Q. and L.Y. wrote the manuscript.

## Additional information

**Competing interests:** The authors declare no competing financial interests.

