## [Peer Review File · Nature Communications]

Reviewers' comments:

Reviewer #1 (Remarks to the Author):

This manuscript by Cong Qiu et al investigated the role of SENP1-mediated GATA2 de-SUMOylation in regulating graft arteriosclerosis. The authors found the endothelial specific induction of SENP1 in clinical graft rejection specimens, and showed that a better vascular reactivity with limited leukocyte recruitment and neointima formation in SENP1-eCKO graft. The authors also showed the increase of GATA2 SUMOylation in graft endothelium from SENP1-eCKO mice, and SUMO fused GATA2 inhibited TNF- α -induced adhesion molecule expression. The data are potentially interesting, but the role of SENP1-mediated GATA2 de-SUMOylation in graft arteriosclerosis was not fully established. In addition, several staining data were confusing and difficult to interpret.

Major

1. In Fig.1 the authors showed the endothelial surface at macro-vessel level in non-diseased and mildly rejecting graft, but the panel of severely rejecting graft showed mid-sized level vessel, probably buried in side the lesion. It is impossible to compare the molecule expression in the different size of vessels. Also please add the control bar (size) in each panel.
2. The authors proposed the role of SENP1 in GATA2 de-SUMOylation. GATA2 mainly expressed in the nucleus. The GATA2 staining shown in mildly rejecting graft is understandable, but the GATA2 staining shown in severely rejecting samples was mainly in the cytosol. This is very confusing.
3. In addition, SENP1 staining shown in severely rejecting graft in Fig.1A was mainly in the cytosol, too. Did the authors think the cytosol translocation of SENP1 in severely rejecting graft? If so, how can extra-nuclear SENP1 de-SUMOylate GATA2 in the nucleus? It is important to show the specificity of anti-SENP1 antibody for immunostaining, especially in the rejecting grafts, for example by using both IgG control and blocking by antigen-peptide.
4. The authors showed that both SENP1 and GATA2 can regulate adhesion molecule expression including ICAM-1, VCAM-1, and P-selectin. Previously, it has been reported that GATA family could not regulate TNF-induced ICAM-1 expression in endothelial cells¹. Please explain about this discrepancy.
5. It has been well established that endothelial cell adhesion molecule expression is mainly regulated by NF- κ B. The authors group and collaborator reported that SENP1-mediated de-SUMOylation inhibits inflammatory response including NF- κ B activation in adipocyte². The authors need to detect NF- κ B activation and NEMO SUMOylation in SENP1 deleted endothelial cells, and explain the relationship between NF- κ B and GATA2 SUMOylation in endothelial cell adhesion molecule induction.
6. In Fig.6F the authors showed that IL-1 β decreased GATA2 SUMOylation in vitro. However, in Fig. 5A there was no difference of the level of GATA2 SUMOylation between basal and graft samples in wild type in vivo. This data makes the patho-physiological role of GATA2 de-SUMOylation in graft arteriosclerosis very unclear, because without SENP1 deletion GATA2 de-SUMOylation was not happened in the natural condition (wild type).
7. In Fig.9A the authors showed that SUMO-fused GATA2 inhibited TNF-induced ICAM-1, VCAM-1, and E-selectin expression. First, how the authors fused SUMO (1, 2, or 3?) to GATA2 (N-terminal or C-terminal?) was not clear. Second, based on the structural difference SUMO-fused GATA2 may not be representative to actual SUMOylated GATA2. Therefore, it will be important to compare K/R and wild type mutant, by which the structural difference may be minimized. The level of TNF-induced adhesion molecules expression between GATA2-WT and GATA2-2KR expressed cells showed no difference, but this could be due to the time after TNF stimulation (24 hrs). Therefore, it will be important to show the time course of TNF-induced adhesion molecule expression (early phase) and examine the expression level difference in GATA2-WT and GATA2-2KR expressed cells.
8. The authors showed that vaso-reactivity was improved in SENP1-eCKO graft. The endothelial-mediated vaso-reactivity is regulated by 1) eNOS expression and activity, and 2) EC apoptosis.

Therefore, it will be crucial to detect these two events in SENP1-ecKO graft. Especially, the role of SENP1 in regulating apoptosis in endothelial cells is controversial. Li et al have reported that SENP1 enhanced ASK-1-dependent apoptosis³. In contrast, Xu et al have reported that SENP1 is protective against hypoxia-induced endothelial apoptosis⁴. Since both HIPK and HIF1a have been reported as the main SENP1 substrate in endothelial cells, it is important to detect HIF1a and HIPK1 SUMOylation and EC apoptosis in graft arteriosclerosis model.

Ref

1. Umetani, M., et al. Function of GATA transcription factors in induction of endothelial vascular cell adhesion molecule-1 by tumor necrosis factor-alpha. *Arterioscler Thromb Vasc Biol* 21, 917-922 (2001).
2. Shao, L., et al. SENP1-mediated NEMO deSUMOylation in adipocytes limits inflammatory responses and type-1 diabetes progression. *Nat Commun* 6, 8917 (2015).
3. Li, X., et al. SENP1 mediates TNF-induced desumoylation and cytoplasmic translocation of HIPK1 to enhance ASK1-dependent apoptosis. *Cell Death Differ* 15, 739-750 (2008).
4. Xu, Y., et al. Induction of SENP1 in endothelial cells contributes to hypoxia-driven VEGF expression and angiogenesis. *J Biol Chem* 285, 36682-36688 (2010).

Reviewer #2 (Remarks to the Author):

Qiu et al identify an important role for SENP1 in the regulation of endothelial cell activation and graft atherosclerosis. They find that sumoylation of GATA2 destabilizes the protein and reduces its ability to bind DNA and activate target genes. SENP1 promotes endothelial activation by antagonizing GATA2 sumoylation. While the studies are novel and interesting, several questions remain:

Major comments:

- 1) The authors refer to GATA2 as the master transcription factor that regulates adhesion molecule expression. What is the evidence that GATA2 is more important in the regulation of adhesion molecules than the well-established role of NF-kB? Have the authors tested whether SENP1 affects NF-kB signaling, which is critical for endothelial activation? Do GATA2 and NF-kB synergize at these promoters? Have any studies assessed GATA2 deletion in the endothelium on graft atherosclerosis? In vivo data directly linking GATA2 to graft atherosclerosis would be helpful.
- 2) The data on EndoMT are not strong since they are based solely on gene expression rather than lineage tracking. The language needs to be toned down on this data.
- 3) In Fig. 2, it would be very helpful to have a non-transplanted control for comparison. Does deletion of SENP1 from the endothelium return neointimal growth to non-transplanted levels? Without this comparison it is difficult to gauge the magnitude of the effect on graft atherosclerosis.
- 4) The authors nicely show that SENP1 expression is affected in human coronary arteries with graft atherosclerosis. What does SENP1 expression look like in mouse graft atherosclerosis? This is crucial data. Showing SENP1 expression in the EC-specific knock-out would also be helpful.
- 5) In Fig. 6B, is this western of endogenous GATA2 or transfected tagged GATA2? The labeling is not clear. A similar comment could be made regarding Fig. 6D. Do these cells have no endogenous SENP1, or is the blot just probing tagged protein?
- 6) In Fig. 8B, an IgG control is required, and the measurement of GATA2 recruitment in unstimulated cells would be helpful. How much is GATA2 recruitment increased in response to TNF-alpha stimulation?

Minor comments:

- 1) The writing needs some improvement to enhance readability.

- 2) It is not clear that the donors and recipients are C57Bl/6 mice. Please clarify this in the methods.
- 3) On p. 20 the authors mention that the clinical data indicates the modulating effect of endothelial SENP1 on GATA2 expression. This is correlative data and does not indicate cause and effect. The language regarding indications of causality should be toned down here.
- 4) The SENP1-CA nomenclature is distracting since 'CA' most often refers to constitutively active, yet this is a non-functional mutant. I realize the authors have used this nomenclature in prior publications, but I think for readability a different abbreviation would be better.
- 5) 'Lumen' is spelled incorrectly in Fig. 2C.

Reviewers' comments:

Reviewer #1 (Remarks to the Author):

This manuscript by Cong Qiu et al investigated the role of SENP1-mediated GATA2 de-SUMOylation in regulating graft arteriosclerosis. The authors found the endothelial specific induction of SENP1 in clinical graft rejection specimens, and showed that a better vascular reactivity with limited leukocyte recruitment and neointima formation in SENP1-eCKO graft. The authors also showed the increase of GATA2 SUMOylation in graft endothelium from SENP1-eCKO mice, and SUMO fused GATA2 inhibited TNF- α -induced adhesion molecule expression. The data are potentially interesting, but the role of SENP1-mediated GATA2 de-SUMOylation in graft arteriosclerosis was not fully established. In addition, several staining data were confusing and difficult to interpret.

Major

1. In Fig.1 the authors showed the endothelial surface at macro-vessel level in non-diseased and mildly rejecting graft, but the panel of severely rejecting graft showed mid-sized level vessel, probably buried inside the lesion. It is impossible to compare the molecule expression in the different size of vessels. Also please add the control bar (size) in each panel.

Reply: We appreciate the comments. Actually, these graft arteries are in the similar size, which can be identified clearly in the low power images in Fig. S1A. The expression of target molecules is more discernable in high power images in Fig.1. But the dramatically reduced lumen resulted from tremendous intimal hyperplasia in severely rejecting graft may make readers confused comparing to the non-diseased and mildly rejecting graft. To clarify this point, we have added control bar for each panel as suggested and descriptive information in the "Results" on p. 17 and figure legend of Figure 1 of the revised manuscript.

2. The authors proposed the role of SENP1 in GATA2 de-SUMOylation. GATA2 mainly expressed in the nucleus. The GATA2 staining shown in mildly rejecting graft is

understandable, but the GATA2 staining shown in severely rejecting samples was mainly in the cytosol. This is very confusing.

Reply: We do agree that GATA2 is mainly expressed in the nucleus. We modified our immunostaining protocol and re-stained GATA2 in severely rejecting graft with controls using both isotype control and antigen blocking peptide. New images are shown in Fig. 1D. The results show that GATA2 expression is mostly in the nucleus and obviously enhanced in luminal endothelial layer of rejecting arteries comparing to non-diseased group. The negative staining in control groups using isotype and antigen blocking peptide excludes the non-specific staining of GATA2 herein (Fig. 1D).

3. In addition, SENP1 staining shown in severely rejecting graft in Fig.1A was mainly in the cytosol, too. Did the authors think the cytosol translocation of SENP1 in severely rejecting graft? If so, how can extra-nuclear SENP1 de-SUMOylate GATA2 in the nucleus? It is important to show the specificity of anti-SENP1 antibody for immunostaining, especially in the rejecting grafts, for example by using both IgG control and blocking by antigen-peptide.

Reply: Also, we re-stained SENP1 in severely rejecting graft with controls using both isotype control and antigen blocking peptide to identify specificity of anti-SENP1 antibody. New images are shown in Fig. 1A. The results show that SENP1 expression is mainly in the nucleus and dramatically enhanced in luminal endothelial layer of rejecting arteries comparing to non-diseased group. The negative results in control groups ensure the specificity of SENP1 staining (Fig. 1A). Therefore, the immunofluorescence results indicate that SENP1 deSUMOylates GATA2 in the nucleus of graft endothelial cell. Correlated modification of immunofluorescence procedure is also made in the "Methods" on p. 10 of the revised manuscript. We thank a lot for the comments for the immunofluorescence in clinical samples.

4. The authors showed that both SENP1 and GATA2 could regulate adhesion molecule expression including ICAM-1, VCAM-1, and, P-selectin. Previously, it has been reported that GATA family could not regulate TNF-induced ICAM-1 expression in endothelial cells¹. Please explain about this discrepancy.

Reply: We appreciated the comments for better understanding the detailed regulatory mechanism of GATA2 on adhesion molecules. It was reported that GATA2 and GATA3 could associate with NF- κ B in a transcription factor complex and acts synergistically with NF- κ B in triggering gene transcription². Therefore, upon TNF- α stimulation, GATA2 may not directly bind to ICAM-1 promoter but can be recruited to the ICAM-1 promoter region by associating with NF- κ B in the transcriptional complex. Accordingly, the expression of endothelial adhesion molecules seems to be regulated by the synergistic action of NF- κ B and GATA2. We have incorporated the related statement in the "Discussion" on p. 30 and p. 34 of the revised manuscript.

5. It has been well established that endothelial cell adhesion molecule expression is mainly regulated by NF- κ B. The authors group and collaborator reported that SENP1-mediated

de-SUMOylation inhibits inflammatory response including NF- κ B activation in adipocyte³. The authors need to detect NF- κ B activation and NEMO SUMOylation in SENP1 deleted endothelial cells, and explain the relationship between NF- κ B and GATA2 SUMOylation in endothelial cell adhesion molecule induction.

Reply: We appreciated the comments to investigate comprehensive modulatory mechanism in endothelial adhesion molecule expression. Indeed, NF- κ B is another important transcription factor in regulating endothelial adhesion molecular expression. In our previous study, we showed that SENP1-mediated NEMO deSUMOylation inhibits NF- κ B activation in adipocyte³. But the role of SENP1 mediated SUMOylation in endothelial NF- κ B signaling is not determined. To this end, we carried out additional experiments in human umbilical endothelial cells (HUVEC), which show that loss of SENP1 decreases NF- κ B (p65) activation induced by TNF- α (Fig. S13A, additional new figure). Nevertheless, loss of SENP1 in HUVEC enhances I κ B α SUMOylation but not NEMO SUMOylation (Fig. S13B, additional new figure). It has been well established that I κ B α SUMOylation inhibits I κ B α degradation thereby attenuates NF- κ B activation⁴. Thus, be different from SENP1 in adipocyte, endothelial SENP1 deficiency inhibits inflammatory NF- κ B activation by promoting I κ B α SUMOylation. On the other hand, GATA2 acted synergistically with NF- κ B in a transcriptional complex as we discussed above, and regulated endothelial adhesion molecule expression with NF- κ B coordinately⁵. In the current study, we set up the direct link of endothelial GATA2 to GA (Fig. S8, additional new figure) and endothelial adhesion molecule induction in vitro (Fig. 9A and B), which demonstrates the important role of GATA2 in endothelial activation and inflammation as NF- κ B. Taken together, we conclude that SENP1-mediated SUMOylation suppress the activity of both NF- κ B and GATA2 in endothelial cells and therefore the synergistic effect of the two transcriptional factors on endothelial cell adhesion molecule induction. We have modified the working model in Fig. 10 and incorporated related statements in the figure legend on p.52 and the "Discussion" on p. 30 in the revised manuscript. Also, we replace "master transcription factor" with "major transcription factor" for the role of GATA2 in regulating adhesion molecules in the revised manuscript.

6. In Fig.6F the authors showed that IL-1 β decreased GATA2 SUMOylation in vitro. However, in Fig. 5A there was no difference of the level of GATA2 SUMOylation between basal and graft samples in wild type in vivo. This data makes the patho-physiological role of GATA2 de-SUMOylation in graft arteriosclerosis very unclear, because without SENP1 deletion GATA2 de-SUMOylation was not happened in the natural condition (wild type).

Reply: We appreciate the concerns for the patho-physiological role of GATA2 deSUMOylation in graft arteriosclerosis. The immunoprecipitation assay in Fig. 6F was performed employing human umbilical endothelial cells, in which proinflammatory cytokine IL-1 β decreased GATA2 SUMOylation in vitro. The minor difference of the level of GATA2 SUMOylation between basal and graft samples in wild type in Fig. 5A might be due to low percentage of endothelial cell number in the whole graft. To determine the patho-physiological role of GATA2 SUMOylation in graft arteriosclerosis in vivo, we did immunofluorescence staining of GATA2 and SUMO1 in non-transplanted donor vessel (defined as "basal") and graft harvested 3 days after

transplantation, both of which are from the same set of **WT** samples employed in Fig.5. As shown in Fig. S9 (additional new figure), decreased SUMOylated GATA2 in endothelial layer was detected in graft compared with basal group, which preserves the truth of endothelial GATA2 deSUMOylation in the natural patho-physiological condition during graft arteriosclerosis. In addition, endothelial SENP1 induction was observed both in WT mice grafts (Fig. S3, additional new figure) and human rejecting arteries (Fig. 1A), indicating the occurrence of natural deSUMOylation in the endothelium within graft arteriosclerosis progression. The above results in Fig. S9 also demonstrate enhanced endothelial GATA2 expression in graft compared with basal group, which further confirms that GATA2 SUMOylation attenuates GATA2 stability as we concluded before. Correlated description was added in the “Results” on p. 23 of the revised manuscript.

7. In Fig.9A the authors showed that SUMO-fused GATA2 inhibited TNF-induced ICAM-1, VCAM-1, and E-selectin expression. First, how the authors fused SUMO (1, 2, or 3?) to GATA2 (N-terminal or C-terminal?) was not clear. Second, based on the structural difference SUMO-fused GATA2 may not be representative to actual SUMOylated GATA2. Therefore, it will be important to compare K/R and wild type mutant, by which the structural difference may be minimized. The level of TNF-induced adhesion molecules expression between GATA2-WT and GATA2-2KR expressed cells showed no difference, but this could be due to the time after TNF stimulation (24 hrs). Therefore, it will be important to show the time course of TNF-induced adhesion molecule expression (early phase) and examine the expression level difference in GATA2-WT and GATA2-2KR expressed cells.

Reply:

-1) The SUMO-fused GATA2 was constructed by fusing SUMO1 to GATA2 (N-terminal) with the same strategy as we employed in the previous study⁶. Briefly, the PCR-amplified SUMO1 fragment (without double-glycine-encoding sequence at the C-terminal) was subcloned into pXF6F vector plasmid using restriction sites of BglII and NotI, followed by subcloning the PCR-amplified human GATA2 cDNA using restriction sites XhoI and XbaI at the C-terminal of SUMO1 fragment. A schematic diagram is included in Fig. S12A (additional new figure) and correlated description was added in the “Methods” on p. 12 of the revised manuscript.

-2) We appreciate the suggestion to focus on the effect of GATA2-KR on endothelial adhesion molecules induction in early phase. Accordingly, HUVEC cells overexpressing GATA2-WT or GATA2-2KR were treated with a short time course (up to 6h) of TNF- α . TNF- α -induced adhesion molecule expression in early phase was determined by quantitative real time PCR. The results showed significant increase of endothelial adhesion molecule induction in GATA2-2KR group compared with GATA2-WT group, which confirms the critical role of GATA2 SUMOylation in endothelial activation (Fig. 9B, additional new figure). Correlated description was added in the “Results” on p. 28 of the revised manuscript.

8. The authors showed that vaso-reactivity was improved in SENP1-ecKO graft. The endothelial-mediated vaso-reactivity is regulated by 1) eNOS expression and activity, and 2) EC apoptosis. Therefore, it will be crucial to detect these two events in SENP1-ecKO graft.

Especially, the role of SENP1 in regulating apoptosis in endothelial cells is controversial. Li et al have reported that SENP1 enhanced ASK-1-dependent apoptosis⁷. In contrast, Xu et al have reported that SENP1 is protective against hypoxia-induced endothelial apoptosis⁸. Since both HIPK and HIF1a have been reported as the main SENP1 substrate in endothelial cells, it is important to detect HIF1a and HIPK1 SUMOylation and EC apoptosis in graft arteriosclerosis model.

Reply:

-1) To determine the effect of endothelial SENP1 deficiency on eNOS in graft aortas, we included additional vaso-reactivity results of myograph analysis in grafts. In this analysis, graft rings were incubated with a NOS inhibitor L-NAME to remove the basal release of eNOS-derived NO followed by a contraction with PE. Although SENP1-ecKO grafts were less responsive to PE, L-NAME caused similar increased isometric tension in both WT and SENP1-ecKO grafts (Fig. S6C, additional new figure). Therefore, the protective effect of endothelial SENP1 deficiency in graft vasomotor dysfunction is mainly attributed to preserved eNOS activity. Correlated description was added in the "Results" on p. 21 and "Methods" on p. 11 of the revised manuscript.

-2) Also, we examined the endothelial apoptosis in the same batch of grafts. In WT and SENP1-ecKO grafts, similar amount of TUNEL-positive endothelial cells were detected (reviewer figure. 1B-C), suggesting similar EC apoptosis in both grafts. In addition, more proliferated endothelial cells were observed by pulse BrdU labeling in the grafts without difference between the two groups (reviewer figure. 1D-E). These results indicate that abundant endothelial proliferation may somewhat compensate the loss of apoptotic endothelial cells in the graft aortas. On the other hand, we examined SUMOylated HIPK1 and HIF1 α in the two groups of grafts by immunoprecipitation. Deficiency of endothelial SENP1 has no effect on HIPK1 and HIF1 α SUMOylation in graft (reviewer figure. 1A), which is consistent with no difference of endothelial cell apoptosis between the two groups of grafts. Therefore, endothelial SENP1 does not play a critical role in endothelial cell apoptosis in graft arteriosclerosis. The graft vessel maintains an intact endothelial layer that develops endothelial activation.

Taken together, we conclude that endothelial SENP1 deficiency improves vaso-reactivity in graft by preserving eNOS activity and thereby endothelial function. These results further support the critical role of SENP1 in regulating EC activation and consequent EC dysfunction.

Reviewer Figure. 1 **Endothelial SENP1 negatively regulates EC apoptosis in graft.** (A) SENP1 deficiency has no effect on HIF1 α and HIPK1 SUMOylation in graft. (B, C) Endothelial SENP1 negatively regulates EC apoptosis. (D, E) C57/BL recipients with WT or SENP1-ecKO donor grafts were injected with BrdU for determining EC proliferation. Data are mean \pm SEM from at least five mice per group followed by unpaired *t* test. N.S = non significance.

Reviewer #2 (Remarks to the Author):

Qiu et al identify an important role for SENP1 in the regulation of endothelial cell activation and graft atherosclerosis. They find that sumoylation of GATA2 destabilizes the protein and reduces its ability to bind DNA and activate target genes. SENP1 promotes endothelial activation by antagonizing GATA2 sumoylation. While the studies are novel and interesting, several questions remain:

Major comments:

1) The authors refer to GATA2 as the master transcription factor that regulates adhesion molecule expression. What is the evidence that GATA2 is more important in the regulation of adhesion molecules than the well-established role of NF- κ B? Have the authors tested whether SENP1 affects NF- κ B signaling, which is critical for endothelial activation? Do GATA2 and NF- κ B synergize at these promoters? Have any studies assessed GATA2 deletion in the endothelium on graft atherosclerosis? In vivo data directly linking GATA2 to graft atherosclerosis would be helpful.

Reply: We appreciate the comments for better understanding the comprehensive modulation of SENP1 in endothelial activation, regarding its effect on both GATA2 and NF- κ B. Indeed, NF- κ B is another important transcription factor in regulating endothelial adhesion molecular expression. In our previous study, we showed that SENP1-mediated NEMO deSUMOylation inhibits NF- κ B activation in adipocyte³. But the role of SENP1 mediated SUMOylation in endothelial NF- κ B signaling is not determined. To this end, we carried out additional

experiments in human umbilical endothelial cells (HUVEC), which show that loss of SENP1 decreases NF- κ B (p65) activation induced by TNF- α (Fig. S13A, additional new figure). Nevertheless, loss of SENP1 in HUVEC enhances I κ B α SUMOylation but not NEMO SUMOylation (Fig. S13B, additional new figure). It has been well established that I κ B α SUMOylation inhibits I κ B α degradation thereby attenuates NF- κ B activation⁴. Thus, different from SENP1 in adipocyte, endothelial SENP1 deficiency inhibits inflammatory NF- κ B activation by promoting I κ B α SUMOylation. On the other hand, it was reported that GATA2 acted synergistically with NF- κ B in a transcriptional complex in triggering gene transcription² and regulated endothelial adhesion molecule expression with NF- κ B coordinately⁵. Also, we set up the direct link of endothelial GATA2 to induction of endothelial adhesion molecule in vitro (Fig. 9A-B) and further to GA (Fig. S8, additional figure, as described in the following), which demonstrates the important role of GATA2 in endothelial activation and inflammation as NF- κ B. Taken together, we conclude that SENP1-mediated SUMOylation suppresses the activity of both NF- κ B and GATA2 in EC and therefore the synergistic effect of the two transcriptional factors on endothelial inflammation in GA (Fig. 10). We have modified the working model in Fig. 10 and incorporated related statements in the figure legend on p. 52 and the "Discussion" on p. 30 in the revised manuscript. Also, we replace "master transcription factor" with "major transcription factor" for the role of GATA2 in regulating adhesion molecules in the revised manuscript. To identify whether GATA2 has a direct link to graft arteriosclerosis, lentiviral endothelial specific GATA2 shRNA or control shRNA were injected into the male graft donor mice via jugular vein before transplantation. Still, the donor aortas were transplanted into female recipients, in which the grafts were rejected by alloimmune response. The GATA2 shRNA significantly suppressed the endothelial GATA2 expression in donor aortas (Fig. S8A, additional new figure), which demonstrated reduced neointima formation (Fig. S8, additional new figure) and inflammatory cell infiltration after transplantation (Fig. S8, additional new figure), comparing to the grafts bearing control shRNA. Thus, the endothelial GATA2 expression in graft vessel is directly connected to graft arteriosclerosis development.

2) The data on EndoMT are not strong since they are based solely on gene expression rather than lineage tracking. The language needs to be toned down on this data.

Reply: We have toned down the description on the role of Endo-MT in GA on p. 21 in the revised manuscript.

3) In Fig. 2, it would be very helpful to have a non-transplanted control for comparison. Does deletion of SENP1 from the endothelium return neointimal growth to non-transplanted levels? Without this comparison it is difficult to gauge the magnitude of the effect on graft atherosclerosis.

Reply: We appreciate the suggestion to include basal control for comparison of graft remodeling. Obviously, non-transplanted vessels have regular morphology without any neointima growth (Reviewer figure. 2). But we think the graft in male to male transplantation is the matched control of graft that develops transplanted arteriosclerosis in male to female transplantation in the current study. The male-to-male graft suffers all the suture injuries but

the immune rejection mediated by the male specific H-Y minor histocompatibility antigen. Therefore, we include size- and post-transplantation-period- matched male-to-male graft as basal control to gauge the magnitude of the effect of SENP1 on graft arteriosclerosis (Fig. 2A). Old Fig S3 showing male-to-male graft images is deleted accordingly. Comparing to the male-to-male graft that has no obvious vascular remodeling and intimal hyperplasia, SENP1-ecKO male-to-female graft keeps similar lumen area but has minor neointima formation, while the WT male-to-female graft exhibits reduced lumen area with considerable neointima. Endothelial SENP1 deficiency significantly suppresses the neointima formation and preserves the vascular morphology of the graft, suggesting the critical role of endothelial SENP1 in graft arteriosclerosis. But the single deletion of SENP1 in endothelium cannot completely block graft arteriosclerosis, because other proteins such as SOCS1, let-7, IL-1 α in endothelium and other regulatory molecules in vessel wall beyond endothelium are also involved in the graft arteriosclerosis progression as reported in our and other previous studies.

Review Figure. 2 Morphology of non-transplanted vessel. A segment of male WT or SENP1-ecKO donor thoracic aorta was harvested for non-transplanted control. No graft arteriosclerosis was observed by Elastica–Van Gieson (EVG) staining. Representative low-powered images (bar represents 200 μ m) and high-powered images (bar represents 50 μ m) are shown in the top row and the bottom row, respectively.

4) The authors nicely show that SENP1 expression is affected in human coronary arteries with graft atherosclerosis. What does SENP1 expression look like in mouse graft atherosclerosis? This is crucial data. Showing SENP1 expression in the EC-specific knock-out would also be helpful.

Reply: We appreciate the suggestion for investigating the pattern of endothelial SENP1 expression in mouse graft atherosclerosis progression. We did immunofluorescence staining of SENP1 and PECAM1 (the endothelial marker) in non-transplanted aorta and one-week transplanted graft in the WT male to female mouse aorta transplantation. Increased expression of endothelial SENP1 was detected concomitant with the graft rejection (Fig. S3, additional new figure), which is consistent with the observation in clinical samples (Fig. 1A and 1B) and cultured endothelial cells (Fig. 6E), thus suggesting the critical role of endothelial SENP1 in GA. Correlated description was added in the “Results” of the revised manuscript. On the other hand, SENP1 EC-specific knock-out leads to endothelial SENP1 deficiency both in cultured mouse aorta endothelial cells and aorta. We did additional immunofluorescence staining of SENP1 in thoracic aorta of WT and SENP1-ecKO mice to confirm the knock-out

efficiency (Fig. S2B, additional new figure). Correlated description was added in the “Results” on p. 18 of the revised manuscript.

5) In Fig. 6B, is this western of endogenous GATA2 or transfected tagged GATA2? The labeling is not clear. A similar comment could be made regarding Fig. 6D. Do these cells have no endogenous SENP1, or is the blot just probing tagged protein?

Reply: The Western blots in Fig. 6B and 6D were all performed probing transfected tagged (Flag and HA) proteins in 293T cells with tag antibodies except the blots for β -actin. We are sorry for the misleading labeling and have made modifications in these figures and the figure legends in the revised manuscript.

6) In Fig. 8B, an IgG control is required, and the measurement of GATA2 recruitment in unstimulated cells would be helpful. How much is GATA2 recruitment increased in response to TNF-alpha stimulation?

Reply: We thank for the suggestion that is better for understanding the role of SENP1 in GATA2 recruitment onto promoters of endothelial adhesion molecules. Accordingly, we have done additional ChIP assays in HUVEC with more groups, including IgG control and unstimulated groups as indicated. New data is updated in Fig. 8B. Little enrichment values are detected in IgG control groups, verifying the specificity of the ChIP assay. In response to TNF- α , the association of GATA2 with ICAM-1, VCAM-1 and E-selectin promoters are increased by 11.36 vs. 3.99, 63.86 vs. 6.39 and 7.70 vs. 1.55 folds respectively, comparing with LacZ-expressed cells vs. SENP1-mutant-expressed cells. The fold changes are determined by comparing enrichment value of TNF- α stimulated cells with enrichment value of unstimulated cells. Therefore, SENP1 dysfunction significantly suppresses the GATA2 recruitment onto promoters of endothelial adhesion molecules. We have included the correlated description in “Results” on p. 27 of the revised manuscript.

Minor comments:

1) The writing needs some improvement to enhance readability.

Reply: We have gone through the manuscript and polished the English writing carefully.

2) It is not clear that the donors and recipients are C57Bl/6 mice. Please clarify this in the methods.

Reply: Both SENP1 lox/lox and SENP1-ecKO mice are in C57Bl/6 background. C57Bl/6 mice were employed as donor receiving lentiviral GATA2 shRNA or scramble shRNA. And all the recipient mice are C57Bl/6 mice. We have clarified it in the “Methods” on p. 9 and the “Results” on p. 18 in the revised manuscript.

3) On p. 20 the authors mention that the clinical data indicates the modulating effect of endothelial SENP1 on GATA2 expression. This is correlative data and does not indicate cause and effect. The language regarding indications of causality should be toned down here.

Reply: We have toned down the statement on p. 23 in the revised manuscript.

4) The SENP1-CA nomenclature is distracting since 'CA' most often refers to constitutively active, yet this is a non-functional mutant. I realize the authors have used this nomenclature in prior publications, but I think for readability a different abbreviation would be better.

Reply: We have replaced SENP1-CA with SENP1-Mut all through the paper.

5) 'Lumen' is spelled incorrectly in Fig. 2C.

Reply: We apologize for the typo. It has been corrected.

Reference

1. Umetani, M. *et al.* Function of GATA transcription factors in induction of endothelial vascular cell adhesion molecule-1 by tumor necrosis factor-alpha. *Arteriosclerosis, thrombosis, and vascular biology* **21**, 917-922 (2001).
2. Chen, G.Y., Sakuma, K. & Kannagi, R. Significance of NF-kappaB/GATA axis in tumor necrosis factor-alpha-induced expression of 6-sulfated cell recognition glycans in human T-lymphocytes. *The Journal of biological chemistry* **283**, 34563-34570 (2008).
3. Shao, L. *et al.* SENP1-mediated NEMO deSUMOylation in adipocytes limits inflammatory responses and type-1 diabetes progression. *Nature communications* **6**, 8917 (2015).
4. Desterro, J.M., Rodriguez, M.S. & Hay, R.T. SUMO-1 modification of I kappa B alpha inhibits NF-kappaB activation. *Molecular cell* **2**, 233-239 (1998).
5. Minami, T. & Aird, W.C. Thrombin stimulation of the vascular cell adhesion molecule-1 promoter in endothelial cells is mediated by tandem nuclear factor-kappa B and GATA motifs. *The Journal of biological chemistry* **276**, 47632-47641 (2001).
6. Yu, L. *et al.* SENP1-mediated GATA1 deSUMOylation is critical for definitive erythropoiesis. *The Journal of experimental medicine* **207**, 1183-1195 (2010).
7. Li, X. *et al.* SENP1 mediates TNF-induced desumoylation and cytoplasmic translocation of HIPK1 to enhance ASK1-dependent apoptosis. *Cell death and differentiation* **15**, 739-750 (2008).
8. Xu, Y. *et al.* Induction of SENP1 in endothelial cells contributes to hypoxia-driven VEGF expression and angiogenesis. *The Journal of biological chemistry* **285**, 36682-36688 (2010).

REVIEWERS' COMMENTS:

Reviewer #1 (Remarks to the Author):

NA

Reviewer #2 (Remarks to the Author):

The authors have comprehensively responded to the previous reviews. I have just a couple of minor suggestions regarding the text:

- 1) On line 135 the authors use the word 'unnegligible'. This term is not very informative. Perhaps 'important' would be more appropriate.
- 2) On lines 146-147 the authors state that 'loss of endothelial SENP1 is essential for preventing EC activation'. I would suggest the following change, 'loss of endothelial SENP1 prevents EC activation.'
- 3) On line 482-483 the authors state, '...which preserves the truth of endothelial GATA2 deSUMOylation'. I would suggest the following change, '...which validates that GATA2 deSUMOylation occurs...'
- 4) On line 630-631 the statement 'be different from SENP1 in adipocytes' is awkward and should be removed.

REVIEWERS' COMMENTS:

Reviewer #1 (Remarks to the Author):

NA

Reviewer #2 (Remarks to the Author):

The authors have comprehensively responded to the previous reviews. I have just a couple of minor suggestions regarding the text:

1) On line 135 the authors use the word 'unnegligible'. This term is not very informative. Perhaps 'important' would be more appropriate.

Reply: We have replaced 'important' with 'unnegligible' in this sentence.

2) On lines 146-147 the authors state that 'loss of endothelial SENP1 is essential for preventing EC activation'. I would suggest the following change, 'loss of endothelial SENP1 prevents EC activation.'

Reply: We have modified this sentence following the reviewer's suggestion.

3) On line 482-483 the authors state, '...which preserves the truth of endothelial GATA2 deSUMOylation'. I would suggest the following change, '...which validates that GATA2 deSUMOylation occurs...'

Reply: We have rephrased this sentence following the reviewer's suggestion.

4) On line 630-631 the statement 'be different from SENP1 in adipocytes' is awkward and should be removed.

Reply: We have removed this sentence.